# Arl15 upregulates the TGFβ family signaling by promoting the assembly of the Smad-complex

**Meng Shi[1,2], Hieng Chiong Tie[1], Mahajan Divyanshu[1], Xiuping Sun[1], Yan Zhou[1], Boon Kim Boh[1], Leah A Vardy[2], Lei Lu[1]\***

[1]School of Biological Sciences, Nanyang Technological University, Singapore, Singapore; [2]A*STAR Skin Research Labs (A*SRL), Agency for Science, Technology and Research, Singapore, Singapore

**Summary** The hallmark event of the canonical transforming growth factor β (TGFβ) family signaling is the assembly of the Smad-complex, consisting of the common Smad, Smad4, and phosphorylated receptor-regulated Smads. How the Smad-complex is assembled and regulated is still unclear. Here, we report that active Arl15, an Arf-like small G protein, specifically binds to the MH2 domain of Smad4 and colocalizes with Smad4 at the endolysosome. The binding relieves the autoinhibition of Smad4, which is imposed by the intramolecular interaction between its MH1 and MH2 domains. Activated Smad4 subsequently interacts with phosphorylated receptor-regulated Smads, forming the Smad-complex. Our observations suggest that Smad4 functions as an effector and a GTPase activating protein (GAP) of Arl15. Assembly of the Smad-complex enhances the GAP activity of Smad4 toward Arl15, therefore dissociating Arl15 before the nuclear translocation of the Smad-complex. Our data further demonstrate that Arl15 positively regulates the TGFβ family signaling.

## Editor's evaluation

In their manuscript, Shi et al. provide intriguing evidence for a novel regulatory mechanism of signaling in the transforming growth factor-β pathway and they identify through a two-hybrid screen a novel interactor of Smad4 called Arl15, which is a small G-protein. They demonstrate that Arl15 binds the MH2 domain of Smad4 leading to a release of its autoinhibitory structure which facilitates complex assembly with the R-Smads. An important mechanistic feature of this study is that the authors show these interactions are induced upon transforming growth factor-β stimulation. Finally, consistent with the proposed mechanism the authors show that a limited set of known downstream transcription factor targets of the transforming growth factor-β pathway are dependent upon Arl15.

**\*For correspondence:**
lulei@ntu.edu.sg

**Competing interest:** The authors declare that no competing interests exist.

## Introduction

The transforming growth factor β (TGFβ) family signaling pathway is initiated by the TGFβ family cytokines, consisting of TGFβs, nodals, activins, growth and differentiation factors, and bone morphogenetic proteins (BMPs) (*Derynck and Budi, 2019*; *Massagué, 2012*; *Schmierer and Hill, 2007*; *Wrana, 2013*). The pathway can profoundly affect the development and homeostasis of animal tissue and has long been recognized as one of the most critical contributors to multiple human diseases such as cancers, fibrotic disorders, and cardiovascular diseases (*Goumans and Ten Dijke, 2018*; *Kim et al., 2018*; *Seoane and Gomis, 2017*). The core molecular framework of this pathway was outlined more than a decade ago (*Massagué, 2012*; *Schmierer and Hill, 2007*). In the canonical TGFβ and BMP signaling pathways (hereafter referred to as the TGFβ and BMP signaling pathways), the TGFβ family

cytokines first bind to and activate the type II and I receptor kinases, which phosphorylate receptor-regulated Smads (R-Smads), including TGFβ-specific Smad2 and 3 (hereafter referred to as TGFβ R-Smads) and BMP-specific Smad1, 5 and 8 (hereafter referred to as BMP R-Smads). Smads share a three-domain organization comprising N-terminal MH1 and C-terminal MH2 domains connected by a linker region. The type I receptor kinase phosphorylates the MH2 domain of an R-Smad. After phosphorylation, phospho-R-Smads and Smad4 assemble into a complex, usually a heterotrimer with two phospho-R-Smads and one Smad4, via their MH2 domains (*Chacko et al., 2001*; *Chacko et al., 2004*). Eventually, the Smad-complex translocates to the nucleus and executes genomic actions by chromatin remodeling, transcriptional activation, or repression of responsive genes. The formation of phospho-R-Smad and Smad4 complex, or the Smad-complex, is a central event in the TGFβ family signaling pathway. However, we still do not entirely understand this event at the molecular and cellular level. A fundamental question is what initiates and regulates the assembly of the Smad-complex. Hata et al. previously reported that the self-autoinhibition of Smads regulates the assembly (*Hata et al., 1997*). They found that, in Smad4, the MH1 domain intramolecularly interacts with the MH2 domain, therefore preventing full-length Smad4 from interacting with R-Smads. What relieves the autoinhibition of Smad4 and how the resulting active Smad4 subsequently complexes with R-Smads are currently unknown.

Small G proteins are molecular switches that regulate diverse cellular processes. In the G protein cycle (*Cherfils and Zeghouf, 2013*), the guanine nucleotide exchange factor (GEF) activates an inactive or GDP-bound G protein to become the GTP-loaded or active form, which binds to its effectors and triggers cellular effects. The active G protein requires the GTPase activating protein (GAP) to hydrolyze the bound GTP and terminate its active state, completing the G protein cycle by returning it to its initial GDP-bound or inactive form. Arf-family G proteins comprise Arf, Sar, and Arf-like (Arl) groups (*Donaldson and Jackson, 2011*; *Sztul et al., 2019*). Arf and Sar group members have well-documented cellular roles in recruiting vesicular coats and regulating lipid production. Arl group has the most members (>20), but most of them are poorly studied. They have diverse cellular functions, such as membrane trafficking, organelle positioning, microtubule dynamics, and ciliogenesis (*Donaldson and Jackson, 2011*; *Sztul et al., 2019*).

Arl15 is an uncharacterized Arl group member. It became interesting after a series of genome-wide association studies linked its gene locus to rheumatoid arthritis, multiple metabolic traits, such as body shape, blood lipid level, and magnesium homeostasis, and metabolic diseases, such as type 2 diabetes mellitus, coronary heart disease, and childhood obesity (*Corre et al., 2018*; *Danila et al., 2013*; *Glessner et al., 2010*; *Li et al., 2014*; *Negi et al., 2013*; *Mahajan et al., 2014*; *Richards et al., 2009*; *Ried et al., 2016*; *Sun et al., 2015*; *Willer et al., 2013*). It is unclear whether *ARL15* is the causative gene and how its genetic changes can lead to the diseases above since its molecular and cellular functions are largely unknown. Recent studies suggested that Arl15 might play a role in insulin signaling, adiponectin secretion, and adipogenesis (*Rocha et al., 2017*; *Zhao et al., 2017*). Here, we identified Arl15 as a novel regulator of the TGFβ family signaling. We found that it can directly bind to and activate autoinhibited Smad4 to promote the assembly of the Smad-complex. At the same time, the Smad4-containing complex negatively feedbacks to Arl15 by accelerating the GTP hydrolysis of Arl15. Therefore, our data demonstrate that Smad4 acts as an effector and GAP for small G protein Arl15.

## Results
### Arl15-GTP can directly interact with Smad4

During our systematic study of Arl group small G proteins, we focused on Arl15 due to its genetic implication in human diseases (*Corre et al., 2018*; *Danila et al., 2013*; *Glessner et al., 2010*; *Li et al., 2014*; *Negi et al., 2013*; *Mahajan et al., 2014*; *Richards et al., 2009*; *Ried et al., 2016*; *Sun et al., 2015*; *Willer et al., 2013*). Arl15 is ubiquitously expressed in human tissues, and its orthologs are present in most metazoans (*Figure 1—figure supplement 1a*). According to the conserved guanine nucleotide-binding motifs of small G proteins (*Feig, 1999*; *Sprang, 1997*; *Sztul et al., 2019*), we introduced GTP non-hydrolyzable mutation (GTP-bound or active form mutation), A86L (hereafter referred to as AL; see below for the experimental confirmation), and GDP-bound or inactive form mutation, T46N (hereafter referred to as TN), to Arl15.

We subsequently performed yeast two-hybrid screening to identify potential interacting partners of Arl15. Using Arl15-AL as a bait, we uncovered Smad4 as one of the most robust hits (*Figure 1—source data 3*). Their interaction was confirmed by immunoprecipitation assays. We found that exogenously expressed and C-terminally GFP-tagged Arl15-AL, but not Arl15-TN, pulled down endogenous Smad4 (*Figure 1a*). In contrast, Arl5b, another member of Arl group, showed negative results in either GTP (Q70L) or GDP-mutant (T30N) form (*Shi et al., 2018*). In the reverse immunoprecipitation, endogenous Smad4 pulled down substantially more endogenous Arl15 in the presence of guanosine 5′-[β,γ-imido]triphosphate (GMPPNP), a non-hydrolyzable GTP analog, than GDP (*Figure 1b*). In *Figure 1b*, the weak pull-down band in the GDP panel is likely due to the cellular GTP. Next, we asked if the TGFβ1 treatment regulates the interaction between endogenous Smad4 and Arl15. We found that, under the control shRNA knockdown (GL2-shRNA), TGFβ1 treatment significantly increased the amount of Arl15 immunoprecipitated by endogenous Smad4 (*Figure 1c and d*). Therefore, our finding suggests that TGFβ1 could stimulate the interaction between Smad4 and Arl15 through a currently unknown mechanism.

To explore which domain or region of Smad4 interacts with active Arl15, we prepared GST-fused Smad4 fragments (*Figure 1—figure supplement 1b*) and tested if they pull down Arl15-AL-GFP expressed in cell lysate (*Figure 1e*; *Figure 1—figure supplement 1c*). We found that the Smad4-MH2 domain, but not the MH1 domain, is sufficient to interact with Arl15-GTP. Furthermore, we noticed that the addition of the linker region significantly increased the pull-down of Arl15-GTP by the MH2 domain, as shown in pull-downs by GST-tagged Smad4-linker-MH2 and full-length Smad4 (*Figure 1e and f*), demonstrating that the linker region probably contributes to the interaction too.

Although all MH2 domains share a similarity in sequences and structures, using purified GST-fusion proteins, we found that, while none of these MH2 domains interacted with His-Arl15-TN (*Figure 1g*), only the MH2 domain of Smad4, but not that of Smad1, 2, and 3, directly bound to purified His-Arl15-AL (*Figure 1g and h*). Extending the MH2 domain of Smad3 to include its linker region did not make the resulting chimera, GST-Smad3-linker-MH2, interact with Arl15-AL either (*Figure 1h*). Since MH2 domains of BMP R-Smads, including Smad1, 5, and 8, share almost identical sequences with >90% identity, our findings indicate that Arl15-GTP directly interacts with Smad4, but not TGFβ and BMP R-Smads.

Many interactions between a small G protein and its effectors involve the switch-II region of the small G protein, which undergoes a disorder-to-order transition upon GTP-binding (*Cherfils and Zeghouf, 2013*; *Vetter and Wittinghofer, 2001*). To investigate the role of the switch-II region, we further mutated Arl15-AL by swapping its switch-II region with that of Arl5b, the Arl group small G protein that localizes to the Golgi (*Shi et al., 2018*; *Figure 1—figure supplement 1a*). The resulting mutant, Arl15-AL-sw2, was able to bind to GTP as demonstrated by the GTP-agarose pull-down assay (*Figure 1—figure supplement 1d*), suggesting that the mutant might fold properly. In the subsequent immunoprecipitation assay, we found that Arl15-AL-sw2-GFP failed to interact with co-expressed Myc-Smad4 (*Figure 1i*), confirming the essential role of the switch-II in the interaction between Arl15 and Smad4.

To search for amino acids in the Smad4-MH2 domain required for interacting with Arl15, we screened eight missense cancer mutations within the Smad4-MH2 domain (*Figure 1—source data 3*). Briefly, we prepared GFP-tagged full-length Smad4 with single-point mutations and tested their interactions with Arl15-AL in the GST pull-down assay. We found that mutations, A532D, E538K, and H541Y, but not M447K, D493H, L495P, R496H, and R497H, substantially disrupted the pull-down of GFP-Smad4 by GST-Arl15-AL (*Figure 1j and k*), suggesting that the region from amino acid 532–541 of Smad4 might be involved in the interaction with Arl15-GTP. In summary, our data demonstrate that Arl15-GTP can directly and specifically interact with the MH2 domain of Smad4.

## Arl15-GTP colocalizes with Smad4 at the endolysosome

We found that endogenous and exogenously expressed Arl15 localized to the Golgi (*Figure 2a*; *Figure 2—figure supplement 1a*). Furthermore, we found that both AL and TN mutants localized to the Golgi similar to the wild type (WT) (*Figure 2—figure supplement 1a*). In addition to the Golgi, WT and AL-mutant Arl15-GFP were also detected at the plasma membrane (PM), early endosome (EE), late endosome (LE), and lysosome (*Figure 2b*; *Figure 2—figure supplement 1b*).

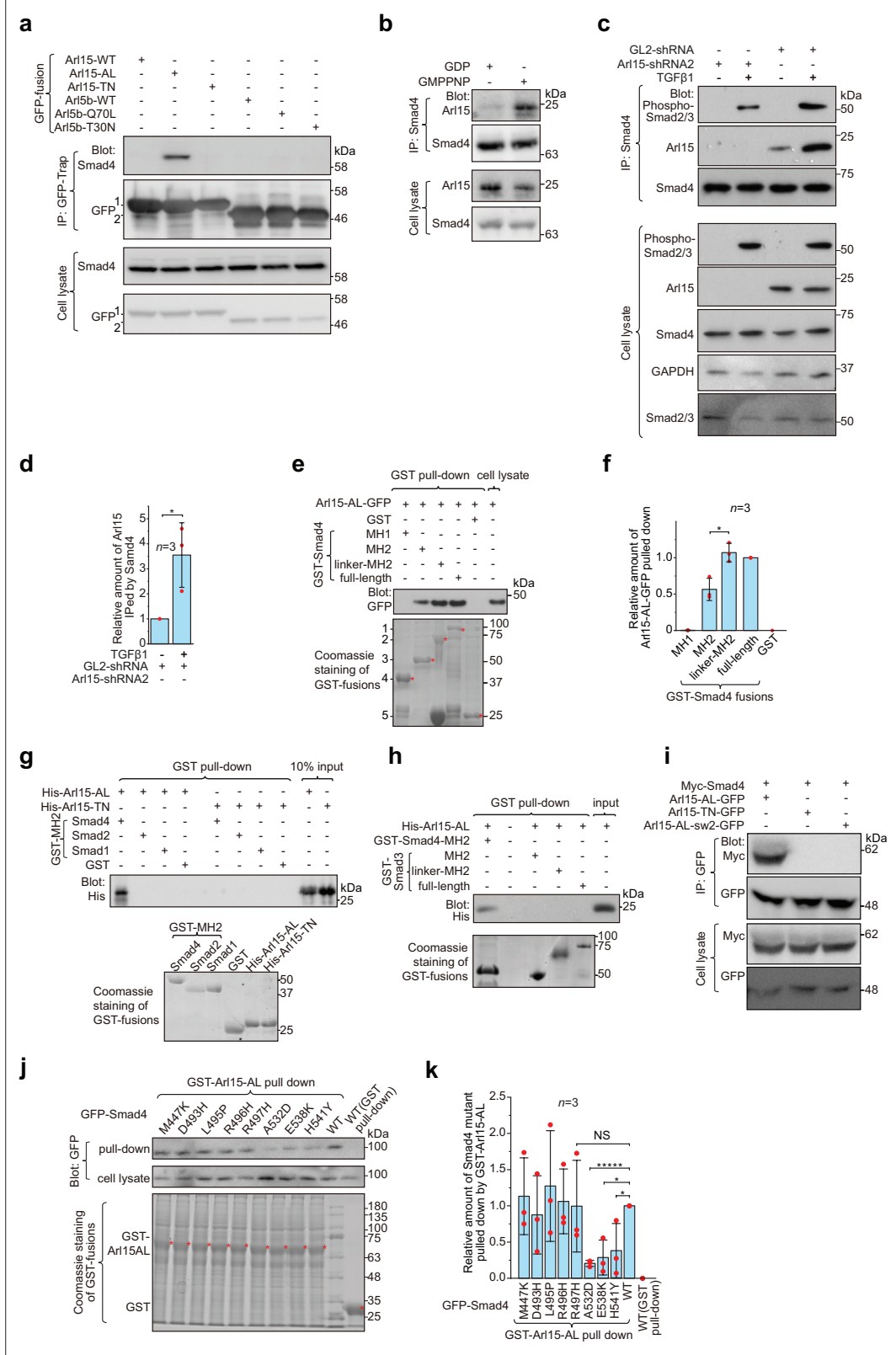

**Figure 1.** Arl15-GTP interacts with Smad4. (**a**) The GTP-mutant form of Arl15 specifically immunoprecipitated endogenous Smad4. HEK293T cell lysates transiently expressing C-terminally GFP-tagged small G proteins were incubated with GFP-Trap beads, and immunoprecipitates were immunoblotted against Smad4 and GFP. 1 and 2 indicate Arl15-(WT, AL, or TN)-GFP and Arl5b-(WT, Q70L, or T30N)-GFP bands, respectively. Arl5b serves as a

*Figure 1 continued on next page*

*Figure 1 continued*

negative control. IP, immunoprecipitation. (**b**) Endogenous Smad4 immunoprecipitated Arl15 in the presence of GMPPNP, but not GDP. HEK293T cell lysates were incubated with anti-Smad4 antibody in the presence of 1 µM GMPPNP or GDP, and immunoprecipitates were immunoblotted against Arl15 and Smad4. (**c, d**) TGFβ1 stimulates the interaction between Smad4 and Arl15, and Arl15 promotes the interaction between Smad4 and phospho-Smad2/3. In (**c**), HEK293T cells subjected to Arl15 or control knockdown were serum-starved for 4 hr followed by further serum starvation or 5 ng/ml TGFβ1 treatment for 20 hr. Cell lysates were incubated with anti-Smad4 antibody, and immunoprecipitates were immunoblotted against indicated antibodies. GAPDH, glyceraldehyde 3-phosphate dehydrogenase. Under the control knockdown, the relative amount of Arl15 immunoprecipitated by Smad4 was quantified in (**d**). To calculate the relative amount of Arl15 immunoprecipitated by Smad4, the band intensity of immunoprecipitated Arl15 is divided by that of corresponding Smad4 and cell lysate Arl15, and the resulting value is further normalized by that of starvation. (**e, f**) The Smad4-MH2 domain specifically pulled down the GTP-mutant form of Arl15. In (**e**), bead-immobilized GST-fusions of Smad4 fragments were incubated with HEK293T cell lysate expressing Arl15-AL-GFP, and pull-downs were immunoblotted against GFP. 1, GST-Smad4; 2, GST-Smad4-linker-MH2; 3, GST-Smad4-MH2; 4, GST-Smad4-MH1; and 5, GST. * indicates specific band. The immunoblot is quantified in (**f**), in which the relative amount of Arl15-AL-GFP pulled down is calculated as the ratio of the intensity of the pull-down band to that of the cell lysate input band. (**g, h**) The MH2 domain of Smad4, but not that of Smad1, 2, and 3, directly interacts with the GTP-mutant form of Arl15. Bead-immobilized GST-fusions of MH2 domains were incubated with purified His-tagged Arl15-AL or TN, and pull-downs were immunoblotted against His-Tag. (**i**) The switch-II region of Arl15 is required for its interaction with Smad4. HEK293T cell lysates expressing indicated proteins were incubated with GFP antibody, and immunoprecipitates were immunoblotted against Myc-tag and GFP. In Arl15-AL-sw2-GFP, the switch-II region of Arl15-AL-GFP is replaced by that of Arl5b. (**j, k**) Cancer missense mutations in the MH2 domain of Smad4 can compromise Arl15-Smad4 interaction. In (**j**), HEK293T cell lysates expressing indicated GFP-Smad4 mutants were incubated with bead-immobilized GST-Arl15-AL, and pull-downs were immunoblotted for GFP. The result was quantified in (**k**). The relative amount of Smad4 mutant pulled down by GST-Arl15-AL is calculated by dividing the band intensity of GFP-Smad4 in pull-down by that in cell lysate, and the resulting value is further normalized by that of GFP-Smad4-WT. In (**e, g, h**, and **j**), the loading of fusion proteins was shown by Coomassie staining. In (**d, f**, and **k**), error bar, mean ± SD of n=3 experiments. p values are from the *t*-test (unpaired and two-tailed). NS, not significant (p>0.05). *, p≤0.05; *****, p≤0.000005. Red dot, individual data point. Molecular weights (in kDa) are labeled in all immunoblots.

The online version of this article includes the following source data and figure supplement(s) for figure 1:

**Source data 1.** Uncropped gel and blot images for *Figure 1*.

**Source data 2.** Numerical data for graphs in *Figure 1d, f and k*.

**Source data 3.** List of positive hits from our yeast two-hybrid screening.

**Source data 4.** List of SMAD4 missense cancer mutations that are tested in GST-Arl15-AL pull-down assay (*Figure 1j and k*).

**Figure supplement 1.** Multiple sequence alignment of Arl15.

**Figure supplement 1—source data 1.** Uncropped gel and blot images for *Figure 1—figure supplement 1*.

Smads likely localize to the endosome since they interact with SARA and endofin, adaptor proteins that possess endosome-targeting FYVE domains (*Chen et al., 2007*; *Gillooly et al., 2001*; *Seet and Hong, 2001*; *Shi et al., 2007*; *Tsukazaki et al., 1998*). Although immunostaining did not reveal a clear membrane association of Smad4, our live-cell confocal imaging uncovered a limited colocalization between mCherry-Smad4 and Arl15-AL-GFP at punctate structures (*Figure 2c*). Notably, the punctate appearance of mCherry-Smad4 was barely discernable against its cytosolic pool. The poor punctate localization might be due to the closed conformation of Smad4, formed by the intramolecular interaction between its MH1 and MH2 domains (see below). Therefore, we tested the mCherry-tagged Smad4-MH2 domain, which does not possess the inhibitory MH1 domain (*Figure 1—figure supplement 1b*). We observed that it displayed a much clearer punctate pattern, which colocalized with Arl15-AL-GFP (*Figure 2d and e*). Our further study revealed that these Smad4-MH2 positive puncta are primarily the EE, LE, and lysosome but not the recycling endosome (RE) (*Figure 2e and f*). Hence, our imaging data suggest that Arl15-GTP might interact with Smad4 at the endolysosome. The absence of the Smad4-MH2 domain at the Golgi, where most Arl15-AL-GFP resides, implies an unknown in vivo mechanism restricting their interaction to the endolysosomal membrane.

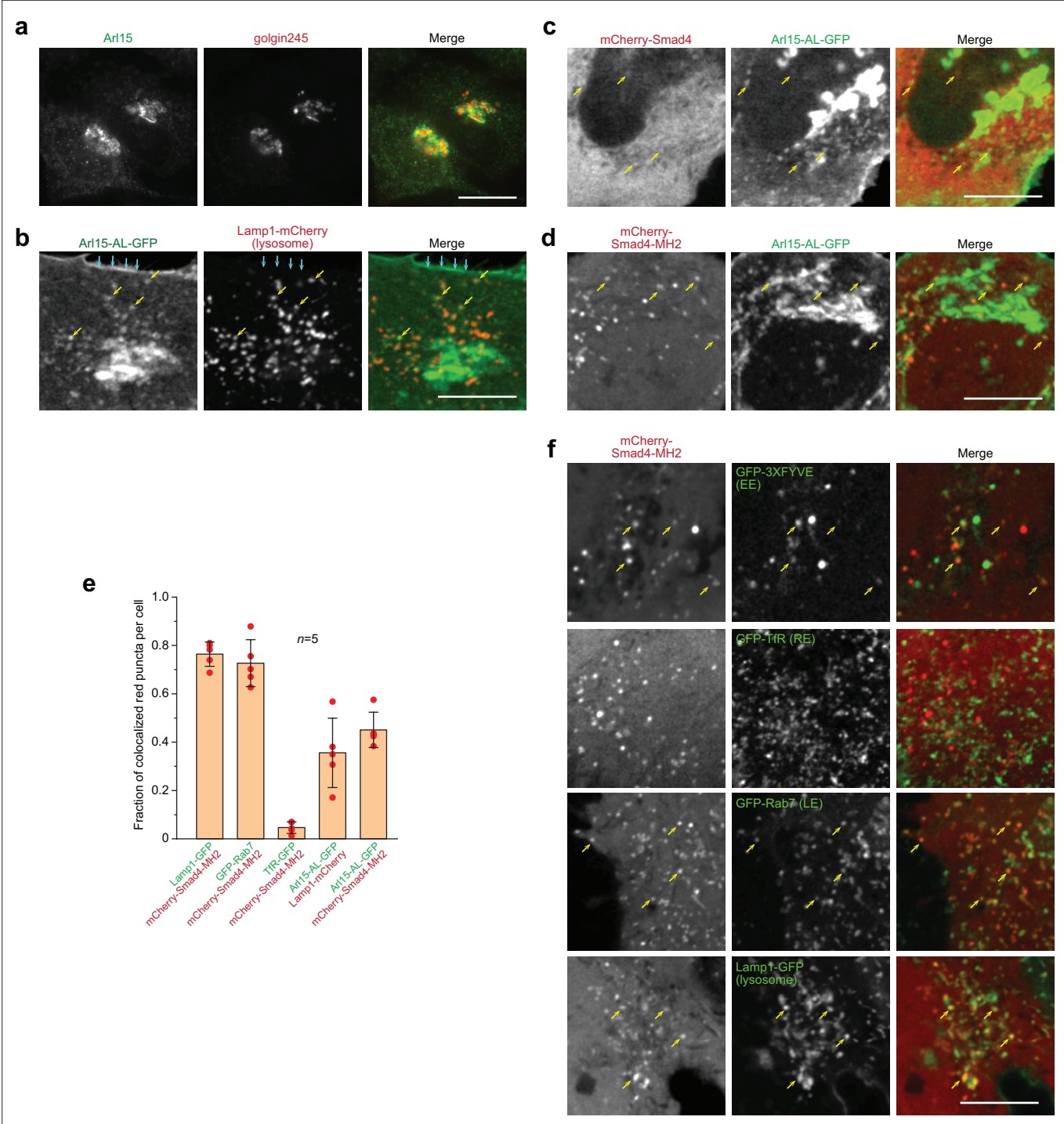

**Figure 2.** Smad4 colocalizes with the GTP-mutant form of Arl15 at the endolysosome. HeLa cells under the normal culture condition were used. (**a, b**) Arl15 localizes to the Golgi and endolysosome. In (**a**), Arl15 localizes to the Golgi. Endogenous Arl15 and golgin245 (a Golgi marker) were stained by immunofluorescence in HeLa cells. Images were acquired by a wide-field microscope. In (**b**), Arl15-AL-GFP localizes to the PM and endolysosome. Live HeLa cells transiently co-expressing Arl15-AL-GFP and Lamp1-mCherry were imaged under a confocal microscope. Yellow arrows, colocalized puncta; cyan arrows, PM. (**c, d**) Smad4-MH2 displays better colocalization with the GTP-mutant form of Arl15 than full-length Smad4 at the endolysosome. Live HeLa cells co-expressing indicated mCherry and GFP-tagged proteins were imaged under a confocal microscope. Note that they colocalize at the endolysosome but not the Golgi. (**e**) Quantitative colocalization between Smad4-MH2 and various endolysosome markers. n=5 cells were imaged, and all red puncta (mCherry-Smad4-MH2 or Lamp1-mCherry) within each image were examined. Fractions of red puncta that visually colocalize with green puncta were calculated and plotted. Error bar, mean ± SD (n=5 cells). Red dot, individual data point. (**f**) Smad4-MH2 localizes to the EE, LE, and lysosome but not the RE. Live HeLa cells co-expressing indicated mCherry and GFP-tagged proteins were imaged under the confocal microscope.

*Figure 2 continued on next page*

*Figure 2 continued*

GFP-tagged 3×FYVE, TfR, Rab7, and Lamp1 are markers for the EE, RE, LE, and lysosome, respectively. Yellow arrows indicate colocalization. Scale bar, 10 μm.

The online version of this article includes the following source data and figure supplement(s) for figure 2:

**Source data 1.** Numerical data for the graph in *Figure 2e*.

**Figure supplement 1.** C-terminally GFP-tagged Arl15 localizes to the Golgi, PM, EE, LE, and lysosome.

## Arl15-GTP indirectly interacts with R-Smads via Smad4

Although Arl15-GTP does not directly bind to R-Smads, we noticed that, in addition to Smad4, a significant amount of endogenous R-Smads, such as Smad2 and Smad1/5/8, was pulled down from cell lysates by GST-Arl15-AL, but not TN (*Figure 3a*). The anti-Smad2/3 antibody used in *Figure 3a* should primarily detect endogenous Smad2 in our HEK293T cells since the band it detected was substantially reduced by the siRNA-mediated knockdown of Smad2 but not Smad3 (*Figure 3—figure supplement 1a*).

Under normal cell culture condition, TGFβs in the serum (*Danielpour et al., 1989*) initiate a basal level of the TGFβ signaling to phosphorylate R-Smads. Since Smad4 can form a complex with phospho-R-Smads via their MH2 domains (*Chacko et al., 2001*; *Chacko et al., 2004*), our finding suggests that Smad4 probably bridges the indirect interaction between Arl15-GTP and phospho-R-Smads. The hypothesis was subsequently confirmed by observing that bead-immobilized GST-Arl15-AL retained substantially more exogenously expressed R-Smads such as Smad1 and 2 (the BMP and TGFβ R-Smad, respectively) in the presence than the absence of co-expressed Smad4 (*Figure 3b and c*; *Figure 3—figure supplement 1b*). In GST-Arl15-AL pull-down without co-expressed Smad4, the residual amount of Smad1 and 2 (indicated by *) was likely due to the indirect interaction mediated by endogenous Smad4. On the other hand, when endogenous Smad4 was depleted by siRNA-mediated knockdown (*Figure 3d and e*), the relative amount of phospho-Smad2/3 immunoprecipitated by Arl15-AL-GFP reduced significantly under TGFβ1 treatment (*Figure 3d and f*).

To avoid endogenous Smad4, we tested similar pull-downs using purified components (*Figure 3g*). To mimic phosphorylated Smad2, we prepared His-Smad2 with S465E/S467E mutations (hereafter referred to as Smad2-SE) (*Liu et al., 1997*). We found that bead-immobilized GST-Arl15-AL pulled down Smad2-SE only in the presence of Smad4, therefore further supporting our hypothesis above. Likely, Arl15-GTP can indirectly interact with other phospho-R-Smads via Smad4 due to high identities shared among MH2 domains of TGFβ or BMP R-Smads. Hence, our data suggest that Arl15-GTP, phospho-R-Smad, and Smad4 might assemble as a complex.

## Arl15-GTP activates Smad4

Smads, including R-Smads and Smad4, adopt a closed conformation by an intramolecular association between MH1 and MH2 domains (*Hata et al., 1997*). Therefore, MH1 inhibits the corresponding MH2 domain and prevents the formation of the Smad-complex. We asked how Arl15-GTP affects the closed conformation of Smad4 by binding to its MH2 domain. Using truncated proteins comprising the MH1 or MH2 domain, we first confirmed that MH1 can interact with the corresponding MH2 domain in Smad4 and Smad2 (*Figure 4a–d*). Furthermore, we observed that co-expressed Arl15-AL, but not Arl15-TN, substantially reduced the interaction between MH1 and MH2 domains of Smad4 (*Figure 4a and b*), demonstrating that Arl15-GTP probably displaces Smad4-MH1 domain by interacting with Smad4-MH2 domain. Hence, our results suggest that active Arl15 might open the closed conformation of Smad4.

It has been documented that the Smad4-MH2 domain can interact with isolated MH2 domains of R-Smads in the absence of C-terminal phosphorylation (*Hata et al., 1997*; *Wu et al., 2001*). We found that the presence of the Smad4-MH2 domain did not substantially reduce the interaction between the MH1 and MH2 domains of Smad2 (*Figure 4c and d*; compare lanes 1 and 4 of the first row of the gel blot). However, further addition of Arl15-AL (lane 2), but not Arl15-TN (lane 3), significantly weakened the interaction, suggesting that Arl15-GTP might promote the Smad4-MH2 domain to engage MH2 domains of R-Smads. Altogether, our biochemical data provide evidence that Arl15-GTP might activate Smad4 by relieving its MH2 domain from the intramolecular inhibition imposed by its MH1 domain.

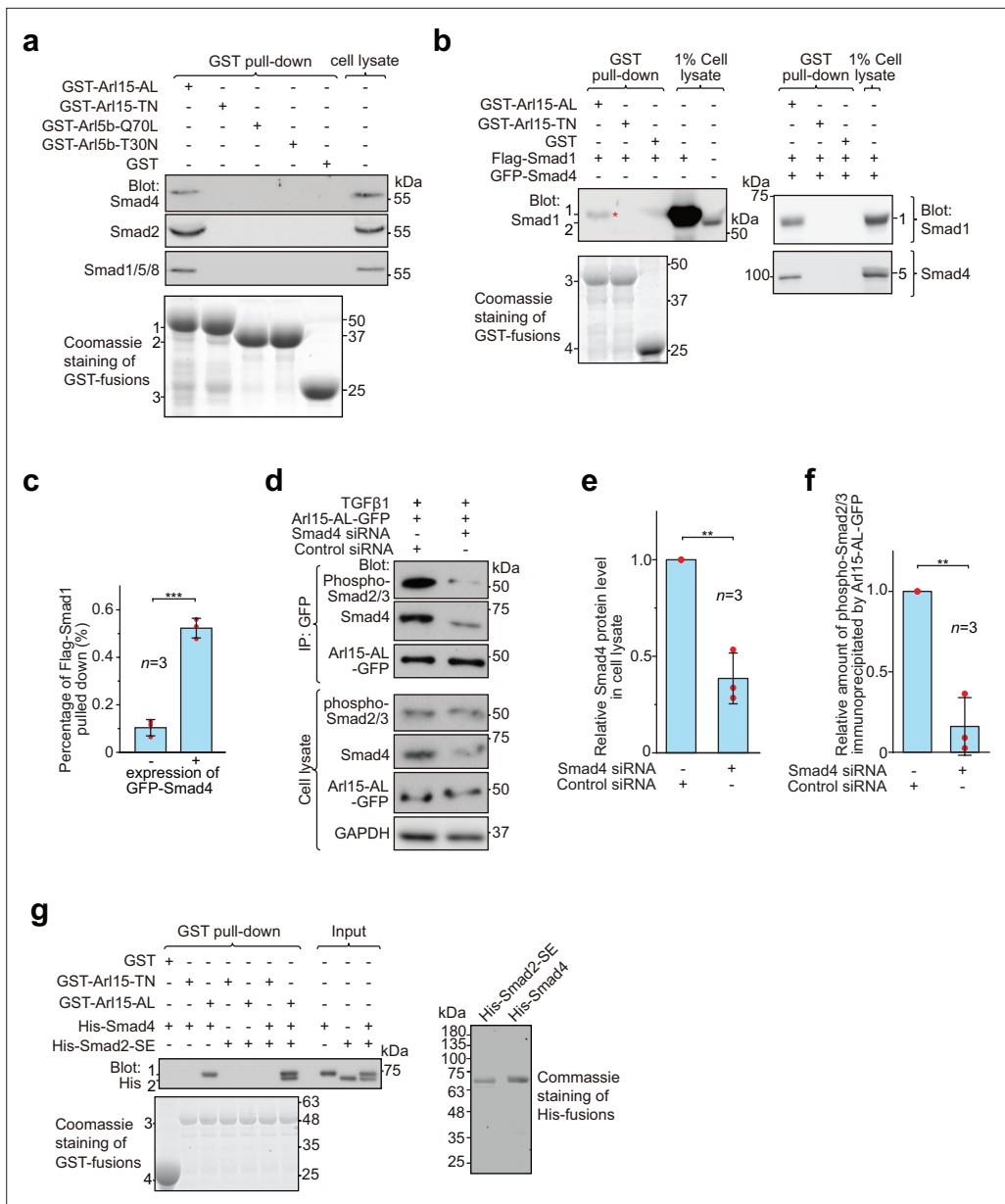

**Figure 3.** Arl15-GTP indirectly interacts with R-Smads via Smad4. HEK293T cells under the normal culture condition were used unless specified. (**a**) The GTP-mutant form of Arl15 specifically pulled down endogenous R-Smads in addition to Smad4. Bead-immobilized GST-fusion proteins were incubated with the cell lysate, and pull-downs and the cell lysate were immunoblotted against indicated Smads. 1, 2, and 3 indicate GST-Arl15 (AL or TN), GST-Arl5b (Q70L or T30N), and GST band. Arl5b serves as a negative control. (**b, c**) The GTP-mutant form of Arl15 pulled down more exogenously expressed Smad1 when Smad4 was co-expressed. In (**b**), bead-immobilized GST-fusion proteins were incubated with cell lysates expressing indicated proteins, and pull-downs and the cell lysates were immunoblotted against Smad1 and 4. 1, Flag-Smad1; 2, endogenous Smad1/5/8; 3, GST-Arl15 (AL or TN); 4, GST; 5, GFP-Smad4; *, the weak band of Flag-Smad1 that was pulled down without co-expression of Smad4. The percentage of Flag-Smad1 pulled down, calculated as the ratio of the intensity of the pull-down band to that of the corresponding 1% cell lysate input band, is plotted in (**c**). (**d, e, f**) The GTP-mutant form of Arl15 immunoprecipitated much less endogenous phospho-Smad2/3 upon Smad4 depletion. In (**d**), cells subjected to siRNA-mediated knockdown were transiently transfected to express Arl15-AL-GFP. After cells were treated with 5 ng/ml TGFβ1 for 20 hr, cell lysates were incubated with the anti-GFP antibody, and immunoprecipitated proteins were blotted together with cell lysates for indicated proteins. IP, immunoprecipitation. The quantification of blots is shown in (**e, f**). To calculate the relative Smad4 protein level in cell lysate, the band intensity of Smad4 in cell lysate is normalized by that of corresponding GAPDH, and the resulting value is further normalized by that of the

*Figure 3 continued on next page*

*Figure 3 continued*

control knockdown. To calculate the relative amount of phospho-Smad2/3 immunoprecipitated by Arl15-AL-GFP, the band intensity of immunoprecipitated phospho-Smad2/3 is normalized by that of corresponding Arl15-AL-GFP and cell lysate phospho-Smad2/3, and the resulting value is further normalized by that of the control knockdown. In (**c**, **e**, and **f**), error bar, mean ± SD of n=3 experiments. p values are from the *t*-test (unpaired and two-tailed). **, p≤0.005; ***, p≤0.0005. Red dot, individual data point. (**g**) Arl15-GTP, Smad4 and Smad2 can assemble into a complex. Bead-immobilized GST-fusion proteins were incubated with indicated purified His-tagged Smads, and pull-downs were immunoblotted against His-tag. 1, His-Smad4; 2, His-Smad2-SE; 3, GST-Arl15 (AL or TN); 4, GST. The loading of fusion proteins is shown by Coomassie staining in (**a**, **b**, and **g**). Molecular weights (in kDa) are labeled in all immunoblots and gels.

The online version of this article includes the following source data and figure supplement(s) for figure 3:

**Source data 1.** Uncropped gel and blot images for *Figure 3*.

**Source data 2.** Numerical data for graphs in *Figure 3c, e and f*.

**Figure supplement 1.** Arl15-GTP indirectly interacts with Smad2 via Smad4.

**Figure supplement 1—source data 1.** Uncropped gel and blot images for *Figure 3—figure supplement 1*.

## Arl15-GTP promotes the assembly of the Smad-complex

We next investigated the effect of Arl15-GTP on the assembly of the Smad-complex in the context of exogenously expressed full-length Smads. In contrast to isolated MH2 domains, full-length R-Smads interact with Smad4 and assemble into a complex only after their C-termini are phosphorylated (*Hata et al., 1997*; *Kretzschmar et al., 1997*). Our data confirmed these reports and further revealed a molecular role of Arl15 in the assembly of the Smad-complex. First, bead-immobilized GST-Smad4 pulled down a substantial amount of Smad2-SE, but not WT and the non-phosphorylatable mutant, Smad2-S465A/S467A (hereafter referred to as Smad2-SA) (*Figure 4e*); consistently, GFP-tagged Smad2-SE, but not WT or Smad2-SA, was found to interact with Myc-Smad4 in the co-immunoprecipitation assay (*Figure 4f*). Second, only Smad2-SE, but not Smad2-SA, was pulled down by bead-immobilized Arl15-AL in a Smad4-dependent manner (*Figure 4g*). Third and most importantly, we observed that the interaction between Smad2-SE and Smad4, that is, the formation of the Smad-complex, was substantially enhanced in the presence of Arl15-AL (*Figure 4e and f*). A similar promoting effect of Arl15-AL on the interaction between Smad1, a BMP R-Smad, and Smad4 was also observed (*Figure 4—figure supplement 1*).

To study Arl15's effect on the Smad-complex assembly in the context of endogenous proteins, we investigated the interaction between Smad4 and phospho-Smad2/3 upon the depletion of Arl15 (*Figure 1c*). We first demonstrated that depleting Arl15 does not significantly change the cellular level of phospho-Smad2/3 stimulated by the TGFβ1 treatment (*Figure 4h*). Importantly, consistent with our above overexpression studies, we found that the depletion of Arl15 substantially reduced phospho-Smad2/3 immunoprecipitated by Smad4 (*Figure 4i*), suggesting an essential role of Arl15 in the efficient assembly of the Smad-complex. Collectively, our data showed that Arl15-GTP might promote the assembly of the Smad-complex by binding to and activating the Smad4-MH2 domain.

## The Smad-complex functions as a GAP to inactivate Arl15-GTP

Once assembled in the TGFβ family signaling pathway, the Smad-complex enters the nucleus to initiate genomic actions (*Derynck and Budi, 2019*; *Massagué, 2012*; *Schmierer and Hill, 2007*; *Wrana, 2013*). We then asked if Arl15-GTP co-translocates to the nucleus together with the Smad-complex. In fluorescence imaging, we observed that Arl15 did not localize to the nucleus under the TGFβ1 treatment (*Figure 5a*). In contrast, the nuclear localization of phospho-Smad2/3 increased substantially (*Figure 5a*). We then further confirm our imaging result and explore the role of Arl15 in the nuclear localization of the Smad-complex using the nuclear fractionation assay. Consistent with our imaging, we did not detect Arl15 in the nuclear fraction in starved or TGFβ1 stimulated control knockdown cells (*Figure 5b*). However, in control or Arl15 knockdown cells, nuclear fractions of phospho-Smad2/3 and Smad4 substantially increased under the TGFβ1 treatment compared to the starvation treatment (*Figure 5b*). Similar observations were made for Arl15 and phospho-Smad1/5/8 in BMP2 stimulated cells (*Figure 5—figure supplement 1a*). Our findings are consistent with our current knowledge of the TGFβ family signaling pathway, in which phospho-R-Smads and Smad4 translocate to the nucleus

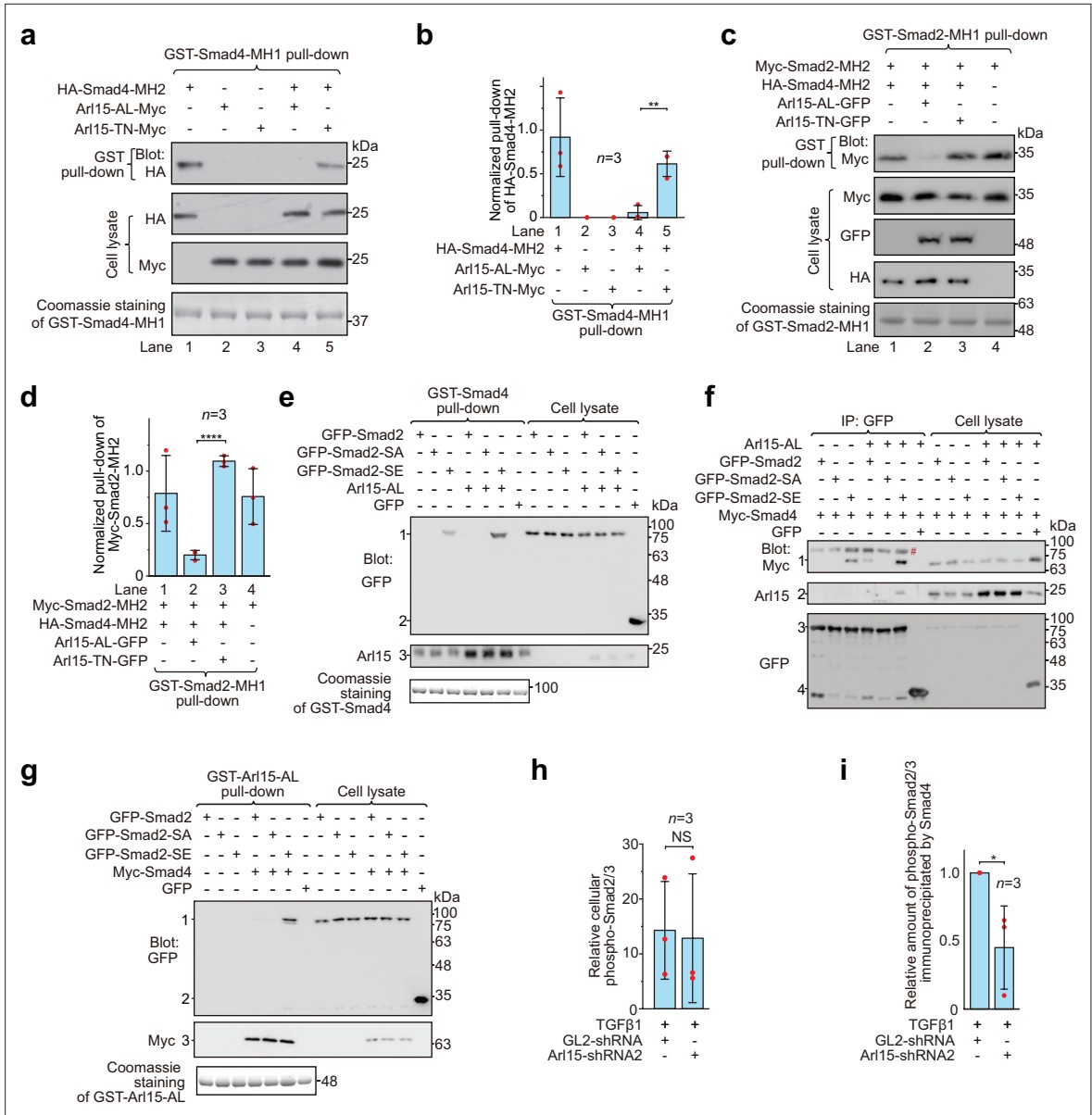

**Figure 4.** Arl15-GTP opens Smad4 and promotes assembly of the Smad-complex. HEK293T cells under the normal culture condition were used. (**a**) The GTP-mutant form of Arl15 opens Smad4 by inhibiting the intramolecular interaction between the MH1 and MH2 domain of Smad4. Bead-immobilized GST-Smad4-MH1 was incubated with the cell lysates expressing indicated proteins, and pull-downs and the cell lysates were immunoblotted against HA or Myc-tag. (**b**) The normalized pull-down of HA-Smad4-MH2 for assays conducted in (**a**). The ratio of the intensity of the pull-down to that of the corresponding cell lysate band was calculated and plotted. (**c**) Arl15-GTP increases the intermolecular interaction between the Smad4-MH2 and Smad2-MH2 domain. Bead-immobilized GST-Smad2-MH1 domain was incubated with the cell lysates expressing indicated proteins, and pull-downs and the cell lysates were immunoblotted against indicated tags. (**d**) The normalized pull-down of Myc-Smad2-MH2 for assays conducted in (**c**). Quantification was the same as in (**b**). (**e, f**) Arl15-GTP promotes the interaction between Smad4 and phosphomimetic mutant of Smad2, Smad2-SE. In (**e**), bead-immobilized GST-Smad4 was incubated with the cell lysates expressing indicated proteins, and pull-downs and the cell lysates were immunoblotted against indicated tags or protein. 1, GFP-Smad2 (WT, SA or SE); 2, GFP; 3, endogenous Arl15 or overexpressed Arl15-AL. In (**f**), the cell lysates expressing indicated proteins were incubated with anti-GFP antibody, and the immunoprecipitates and the cell lysates were immunoblotted against indicated tags or protein. 1, Myc-Smad4; 2, endogenous Arl15 or overexpressed Arl15-AL; 3, GFP-Smad2 (WT, SA or SE); 4, GFP; #, non-specific band. (**g**) Arl15-GTP, Smad4, and Smad2-SE can assemble into a complex. Bead-immobilized GST-Arl15-AL was incubated with the cell lysates expressing indicated proteins, and pull-downs and the cell lysates were blotted. 1, GFP-Smad2 (WT, SA or SE); 2, GFP; 3, Myc-Smad4. In (**a,c,e** and **g**), loading of GST-fusion proteins is shown by Coomassie staining. Molecular weights (in kDa) are labeled in all immunoblots. (**h, i**) Quantification plots of *Figure 1C*, showing that Arl15 promotes the interaction between Smad4 and phospho-Smad2/3. In (**h**), to calculate the relative cellular phospho-Smad2/3, the band intensity of cell lysate phospho-Smad2/3 is normalized by that of corresponding GAPDH. In (**i**), to calculate the relative amount of phospho-

*Figure 4 continued on next page*

*Figure 4 continued*

Smad2/3 immunoprecipitated by Smad4, the band intensity of immunoprecipitated phospho-Smad2/3 is normalized by that of corresponding Smad4 and cell lysate phospho-Smad2/3, and the resulting value is further normalized by that of control knockdown. In (**b**, **d**, **h**, and **i**), error bar, mean ± SD of n=3 experiments. p values are from the *t*-test (unpaired and two-tailed). NS, not significant (p>0.05); *, p≤0.05; **, p≤0.005; ****, p≤0.00005. Red dot, individual data point.

The online version of this article includes the following source data and figure supplement(s) for figure 4:

**Source data 1.** Uncropped gel and blot images for *Figure 4*.

**Source data 2.** Numerical data for graphs in *Figure 4b, d, h and i*.

**Figure supplement 1.** Arl15-GTP promotes the interaction between Smad4 and phosphomimetic mutant of Smad1, Smad1-SE.

**Figure supplement 1—source data 1.** Uncropped gel and blot images for *Figure 4—figure supplement 1*.

upon stimulation by TGFβ family cytokines (*Derynck and Budi, 2019*; *Massagué, 2012*; *Schmierer and Hill, 2007*; *Wrana, 2013*). Notably, when Arl15 was depleted, substantially less phospho-Smad2/3 and Smad4 were found in the nuclear fraction (*Figure 5b–d*). The finding thus supports our hypothesis that Arl15 is essential for the efficient assembly of the Smad-complex, which subsequently translocates to the nucleus.

The absence of Arl15 in the nucleus suggests that Arl15 might dissociate from the Smad-complex before the latter's nuclear translocation. Our findings prompted us to test the hypothesis that the Smad-complex might act as a GAP to inactivate and consequently dissociate Arl15. To that end, we first purified recombinant His-tagged Smad2-SE, Smad4, and Arl15 (WT or AL) (*Figure 5—figure supplement 1b*). Next, His-Arl15 (WT or AL) was first loaded with GTP and subsequently incubated with or without different combinations of His-Smad2-SE and His-Smad4. The inorganic phosphate released during the GTP hydrolysis was enzymatically converted and continuously monitored by spectrophotometry (see Materials and methods). We found that Arl15-WT alone displayed a weak GTP-hydrolysis activity and that the presence of Smad4, but not Smad2-SE, substantially accelerated the GTP hydrolysis rate of Arl15, demonstrating Smad4 as a potential GAP for Arl15 (*Figure 5e*).

Interestingly, the addition of Smad2-SE greatly enhanced the GAP activity of Smad4 toward Arl15. As expected for a small G protein, we observed that AL mutation abolished the GTP hydrolysis activity of Arl15 in all cases — either alone or with the addition of Smad4 or Smad2-SE, retrospectively confirming AL as the GTPase-defective mutation. Using purified domains of Smad4 (*Figure 5—figure supplement 1b*), we mapped the GAP activity to the Smad4-MH2 domain (*Figure 5f*). The extension of the MH2 domain to include the linker region further increased the GAP activity (*Figure 5f*). The positive effect of the linker region on the GAP activity of the MH2 domain is probably due to the enhanced interaction between the Smad4-MH2 domain and Arl15-GTP (*Figure 1e and f*). In summary, our data suggest that, after the assembly of the Smad-complex, Smad4 might have an enhanced GAP activity and consequently inactivate Arl15 and dissociate the latter from the complex.

To better understand the significance of dissociating Arl15 from the Smad-complex, we investigated the effect of overexpressing Arl15-AL on the TGFβ1-stimulated nuclear translocation of the Smad-complex. Since Arl15-AL does not significantly hydrolyze the bound GTP (*Figure 5e and f*), we expect that it could tightly bind to the Smad-complex with minimal dissociation. HeLa cells transfected with Arl15-AL or empty vector (control) were subjected to starvation or TGFβ1-stimulation as *Figure 5b*. Our nuclear fractionation assay demonstrates that Arl15-AL reduced the percentage of nuclear Smad4 and phospho-Smad2/3 compared to the control (*Figure 5g–i*). Therefore, when bound, Arl15 seems to inhibit the nuclear translocation of the Smad-complex. The Smad-complex must inactivate and dissociate the bound Arl15 for efficient nuclear translocation.

## Arl15-GTP is a positive and essential regulator for the full activity of TGFβ and BMP signaling pathways

To understand the cellular significance of Arl15-Smad4 interaction, we investigated the effect of overexpression or depletion of Arl15 on TGFβ-induced transcriptions in HeLa cells. We first tested the transcription of Smad binding element ×4 luc (SBE ×4 luc), a luciferase reporter that comprises four tandem repeats of SBEs as its enhancer (*Zawel et al., 1998*). SBE ×4 luc is commonly used for assaying the TGFβ R-Smad-dependent transcription or TGFβ signaling. We found that when HeLa cells were treated with the serum-free medium (serum-starvation or hereafter referred to as starvation),

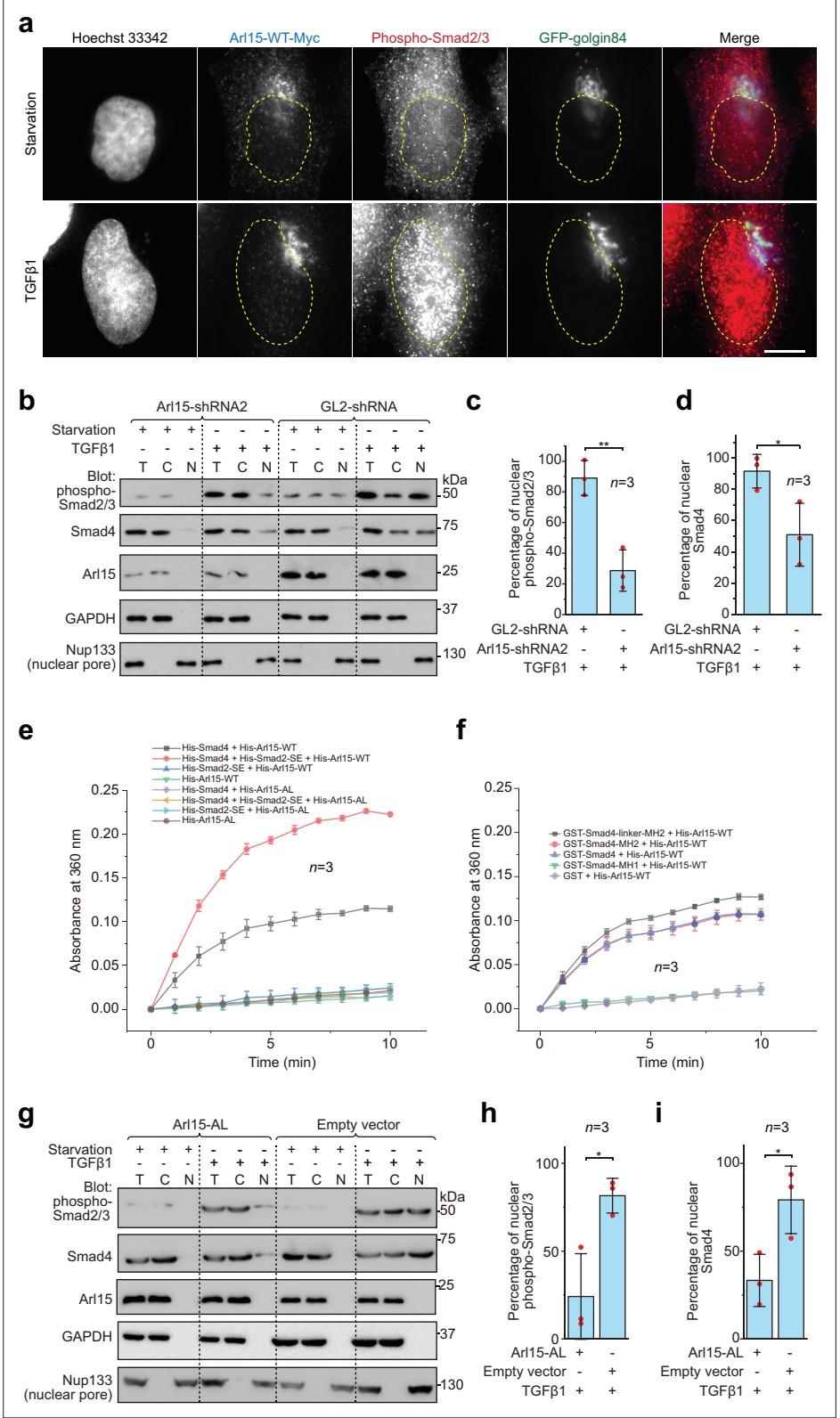

**Figure 5.** The Smad-complex functions as a GAP to inactivate and dissociate Arl15 before translocating to the nucleus. (**a–d**) Under the TGFβ1 treatment, phospho-Smad2/3, but not Arl15, translocated to the nucleus. In (**a**), serum-starved HeLa cells expressing GFP-golgin84 (a Golgi marker) and Arl15-WT-Myc were either further serum-starved or treated with 10 ng/ml TGFβ1 for 1 hr. Cells were stained for DNA (Hoechst 33342)

*Figure 5 continued on next page*

*Figure 5 continued*

and endogenous phospho-Smad2/3. The dotted line indicates the contour of the nucleus. Scale bar, 10 µm. In (**b**), HeLa cells were subjected to Arl15 or control knockdown. After being serum-starved for 4 hr, cells were either further serum-starved or treated with 10 ng/ml TGFβ1 for 1 hr. Total cell lysate (T) and cytosol (C) and nuclear (N) fractions were immunoblotted against indicated proteins. Molecular weights (in kDa) are labeled in immunoblots. Immunoblots were quantified in (**c** and **d**). In (**c**), the band intensity of phospho-Smad2/3 is first normalized by that of corresponding Nup133. To calculate the percentage of nuclear phospho-Smad2/3, the normalized nuclear phospho-Smad2/3 is divided by that of total cell lysate. In (**d**), the percentage of nuclear Smad4 is calculated as in (**c**). Red dot, individual data point. (**e**) Phospho-Smad2 (Smad2-SE) promotes the GAP activity of Smad4 toward Arl15. 40 µM GTP-loaded His-Arl15-WT or AL was incubated with 0.4 µM indicated His-Smads at 22 °C. Released inorganic phosphate was enzymatically converted and continuously monitored by absorbance at 360 nm. The absorbance was plotted against time. (**f**) The Smad4-MH2 domain possesses the GAP activity toward Arl15. The experiment was conducted as in (**e**). GST-fused Smad4 fragments were used. In (**c**, **d**, **e**, and **f**), error bar, mean ± SD of n=3 experiments. (**g, h, i**) Overexpression of GTP-form mutant of Arl15, Arl15-AL, inhibitsTGFβ1-stimulated nuclear translocation of phospho-Smad2/3 and Smad4. HeLa cells transiently expressing Arl15-AL or empty vector control (pCI-neo) were serum-starved for 4 hr followed by further starvation or 5 ng/ml TGFβ1 treatment for 20 hr. Nuclear fractionation, immunoblotting, and subsequent quantification are similar to (**b–d**).

The online version of this article includes the following source data and figure supplement(s) for figure 5:

**Source data 1.** Uncropped gel and blot images for *Figure 5*.

**Source data 2.** Numerical data for graphs in *Figure 5c, d, e, h and i*.

**Figure supplement 1.** Nucleus translocation of phospho-Smad1/5/8 and Coomassie gels of purified proteins used in GAP assays.

**Figure supplement 1—source data 1.** Uncropped gel and blot images for *Figure 5—figure supplement 1*.

overexpression of Arl15-WT or AL, but not TN, was sufficient to stimulate the transcription of SBE ×4 luc to ≥2 fold that of the control (*Figure 6a*). However, the stimulation was much weaker than that of TGFβ1 treatment, which is ~25 fold that of control (*Figure 6a*). The stimulation was abolished by SB431542 (*Figure 6—figure supplement 1a*), a small molecule inhibitor of TGFβ type I receptor kinase activity (*Laping et al., 2002*). A possible explanation for these observations is that Arl15-AL might amplify the autocrine TGFβ signaling in HeLa cells (*Qing et al., 2004*) by promoting the formation of the Smad-complex. Similarly, we found that overexpression of Arl15-WT or AL, but not TN, also stimulated the transcription of BRE-luc (*Figure 6—figure supplement 1b*), a luciferase reporter for assaying the BMP R-Smad-dependent transcription or BMP signaling (*Korchynskyi and ten Dijke, 2002*). On the other hand, when endogenous Arl15 was depleted by RNAi (*Figure 6—figure supplement 1c*), TGFβ1 or BMP2-induced reporter transcription was attenuated to less than half of the control, and the autocrine stimulated transcription also decreased under starvation (*Figure 6b*; *Figure 6—figure supplement 1d*). Hence, our data imply that Arl15 is a positive and essential regulator for the complete activity of TGFβ and BMP signaling pathways.

Since some Arl group small G proteins can regulate intracellular trafficking (*Donaldson and Jackson, 2011*; *Sztul et al., 2019*), we wondered if Arl15 indirectly controls TGFβ family signaling by its role in intracellular trafficking. Therefore, we first investigated if Arl15 is required for secretory trafficking. Using ManII (a Golgi transmembrane glycosidase) and TNFα (a PM-targeted transmembrane protein) RUSH reporters (*Boncompain et al., 2012*), we found that Arl15 knockdown did not substantially affect the ER-to-Golgi (*Figure 6—figure supplement 1e* and f) and the subsequent Golgi-to-PM trafficking (*Figure 6—figure supplement 1g* and h). Next, we observed that TGFβ1-stimulated phosphorylation of Smad2/3 normally occurred upon knockdown of Arl15 (*Figure 1c*, *Figure 4h*, and *Figure 6—figure supplement 1i*). Therefore, Arl15 is probably not required for steps leading to phosphorylation of Smad2/3, such as trafficking and maintenance of TGFβ1 receptors and Smad2/3. In summary, our data argue against the hypothesis that Arl15 indirectly regulates the TGFβ family signaling by intracellular trafficking. Instead, they support a model in which Arl15 regulates the TGFβ family signaling by promoting the assembly of the Smad-complex.

## Arl15 is essential for the efficient transcription of TGFβ target genes

In addition to luciferase reporters, we also studied the effect of disrupting Arl15 on the cellular transcription profile of TGFβ target genes. To that end, we employed the MCF7 cell line, which represents

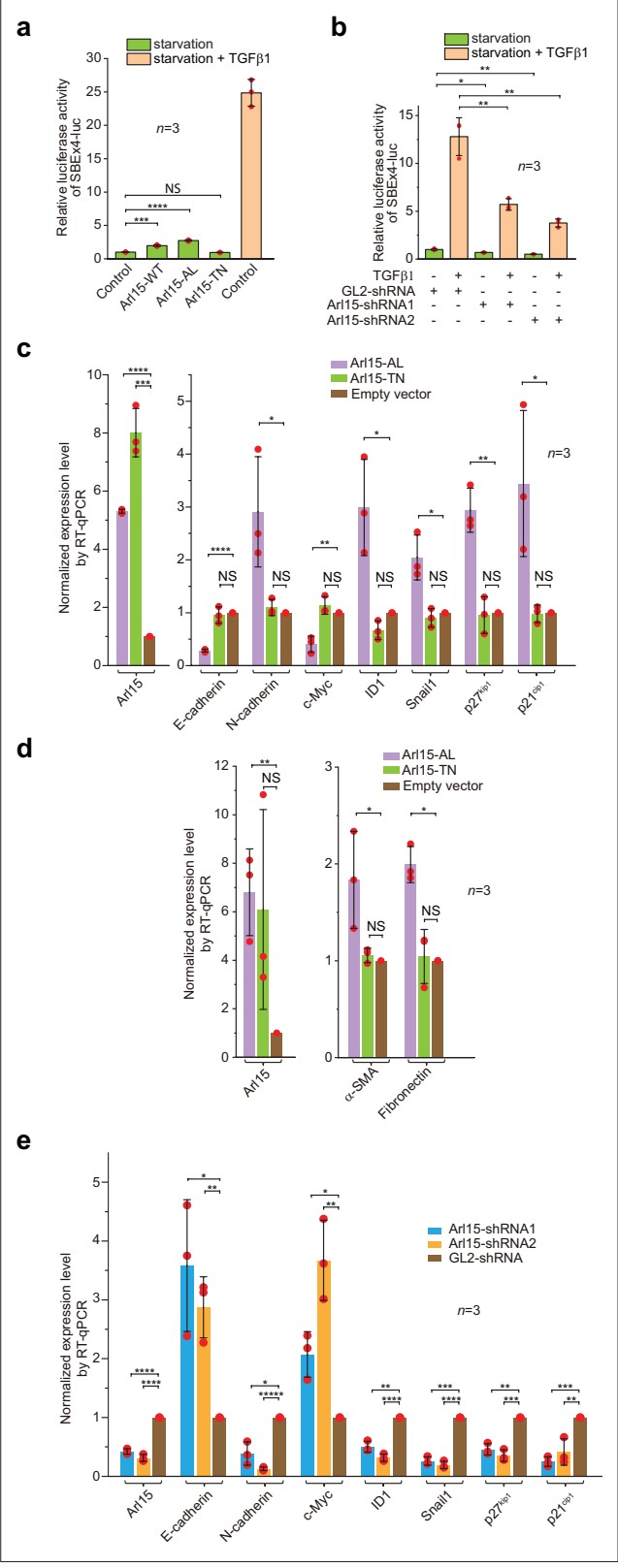

**Figure 6.** Arl15-GTP is a positive and essential regulator for the full activity of TGFβ family signaling pathway.
(**a**) Arl15 can positively regulate the TGFβ signaling pathway since overexpressed Arl15-WT and AL, but not TN, promotes the transcription of SBE ×4 luc reporter under starvation. HeLa cells co-expressing the SBE ×4-driven firefly luciferase and SV40-driven renilla luciferase together with indicated Arl15 mutant or pBluescript SK vector

*Figure 6 continued on next page*

*Figure 6 continued*

DNA (control) were serum-starved for 24 hr. For TGFβ1 induction, control cells were serum-starved for 4 hr followed by 5 ng/ml TGFβ1 treatment for 20 hr. Dual-luciferase assays were performed, and relative luciferase activities were subsequently acquired and normalized. (**b**) Arl15 is essential for efficient TGFβ1 signaling since its depletion reduces TGFβ1-stimulated transcription of SBE ×4 luc reporter. After lentivirus-mediated knockdown of Arl15, HeLa cells co-expressing the dual-luciferase described in (**a**) were serum-starved for 4 hr followed by either further starvation or 5 ng/ml TGFβ1 (starvation +TGFβ1) treatment for 20 hr. Relative luciferase activities were subsequently acquired and normalized. GL2 is a non-targeting control shRNA. (**c, d**) Overexpression of Arl15-AL, but not TN, promotes the transcription of N-cadherin, ID1, Snail1, p27$^{kip1}$, p21$^{cip1}$, α-SMA, and Fibronectin and suppresses the transcription of E-cadherin and c-Myc. MCF7 cells were subjected to lentivirus-transduced overexpression of Arl15-AL or TN followed by starvation for 16 hr. Transcripts of indicated genes were quantified by RT-qPCR and normalized by control (empty vector). (**e**) Opposite to overexpression, depletion of Arl15 suppresses TGFβ1-induced transcription of N-cadherin, ID1, Snail1, p27$^{kip1}$, and p21$^{cip1}$, and promotes the transcription of E-cadherin and c-Myc. After lentivirus-transduced knockdown of Arl15, MCF7 cells were subjected to 5 ng/ml TGFβ1 treatment for 72 hr. Transcripts of indicated genes were quantified and normalized as in (**c**). Error bar, mean ± SD of n=3 experiments. p values were from the *t*-test (unpaired and two-tailed). NS, not significant (p>0.05); *, p≤0.05; **, p≤0.005; ***, p≤0.0005; ****, p≤0.00005; *****, p≤0.000005. Red dot, individual data point.

The online version of this article includes the following source data and figure supplement(s) for figure 6:

**Source data 1.** Numerical data for graphs in *Figure 6a–e*.

**Figure supplement 1.** Arl15 promotes the TGFβ family signaling pathway.

**Figure supplement 1—source data 1.** Uncropped gel and blot images for *Figure 6—figure supplement 1*.

**Figure supplement 1—source data 2.** Numerical data for graphs in *Figure 6—figure supplement 1a, b, d, f, h and j*.

early-stage or pre-malignant breast cancer cells (*Comşa et al., 2015*; *Holliday and Speirs, 2011*). In serum-starved MCF7 cells, overexpression of Arl15-AL, but not TN and empty vector control, was sufficient to upregulate the transcription of N-cadherin, ID1, Snail1, p27$^{kip1}$, p21$^{cip1}$, Fibronectin, and α-SMA, and downregulate the transcription of E-cadherin and c-Myc (*Figure 6c and d*). The increase of p27$^{kip1}$ and p21$^{cip1}$ and decrease of c-Myc transcripts correlate with the cytostatic effect of TGFβ signaling, while the rise of N-cadherin, Snail1, Fibronectin, and α-SMA, and reduction of E-cadherin transcripts characterize the EMT (*Hao et al., 2019*; *Lamouille et al., 2014*; *Seoane and Gomis, 2017*; *Xu et al., 2009*). On the other hand, under TGFβ1 treatment, depletion of Arl15 in MCF7 cells reversed the transcriptional trend of the above genes compared to control knockdown, that is, those that were upregulated by Arl15-AL overexpression (e.g., N-cadherin, ID1, Snail1, p27$^{kip1}$, and p21$^{cip1}$) became downregulated, and vice versa (e.g. E-cadherin and c-Myc) (*Figure 6e*). A similar result was obtained when Arl15 was depleted in the MDA-MB-231 cell line, representing highly metastatic breast cancer cells (*Figure 6—figure supplement 1j and k*). Therefore, our observation correlates the activity of Arl15 with a gene transcription profile characteristic of the TGFβ-induced cytostasis and EMT in cancer cells.

Surprisingly, we found that overexpression of Arl15-AL slightly reduced the transcription of SBE ×4 luc in HeLa cells under TGFβ1 treatment (*Figure 6—figure supplement 1l*). The finding was in contrast to our luciferase assays conducted under starvation, in which overexpression of Arl15-AL upregulates both TGFβ and BMP pathways, although to a limited extent (*Figure 6a*; *Figure 6—figure supplement 1b*). However, our observation supports the inhibitory role of Arl15-AL in the nuclear translocation of the Smad-complex (*Figure 5g–i*). Therefore, we propose that Arl15-AL might have two opposing effects on the Smad-complex – promoting its assembly and inhibiting its nuclear translocation. Under starvation, the assembly of the Smad-complex might be the rate-limiting step when cellular phospho-Smad2/3 concentration is low under autocrine stimulation. Arl15-AL might spontaneously dissociate from the Smad-complex, freeing the Smad-complex for its subsequent nuclear translocation. Hence, the net effect is that Arl15-AL promotes transcription of SBE ×4 luc (*Figure 6a*; *Figure 6—figure supplement 1b*). In contrast, under TGFβ1 treatment, when the cellular phospho-Smad2/3 concentration is high, the nuclear translocation of the Smad-complex becomes the rate-limiting step, and the net effect of Arl15-AL overexpression might be inhibitory.

## Efficient in vitro migration and invasion of cancerous cells require Arl15

The TGFβ signaling pathway is a well-known promoting factor for the metastasis of cancer cells (*Hao et al., 2019*; *Lamouille et al., 2014*; *Seoane and Gomis, 2017*). To explore the role of Arl15 in cancer metastasis, we assessed two key metastatic traits in vitro, cellular migration and invasion, using a highly metastatic breast cancer cell line – MDA-MB-231. By wound healing and collagen gel invasion assays, we observed that the migration and invasion of MDA-MB-231 cells were significantly reduced upon depletion of Arl15 (*Figure 7a–d*). Although the precise role of Arl15 in tumorigenesis requires further investigation, our current data are consistent with what has been documented about the TGFβ signaling pathway in the metastasis of cancers (*Derynck and Budi, 2019*; *Hao et al., 2019*; *Lamouille et al., 2014*; *Massagué, 2012*; *Schmierer and Hill, 2007*; *Seoane and Gomis, 2017*), thus further supporting Arl15 as a positive and essential regulator for the complete activity of this pathway.

## Some Arl15 somatic mutations from cancer patients can compromise Arl15-Smad4 interaction

Using cancer genomic databases, we explored mutations in cancer patients that can disrupt Arl15-Smad4 interaction. We identified and tested somatic missense mutations at the G3-motif or switch-II region of Arl15: E82K, R90L, R95C, and Y96F (*Figure 1—figure supplement 1a* and *Figure 7—source data 4*) with two mutations at other locations, D58N and R150H. When each mutation was introduced in Arl15-AL, the resulting mutant protein displayed normal GTP-binding activity in the GTP-agarose pull-down assay (*Figure 7—figure supplement 1*), suggesting that these mutations might not disrupt the folding of Arl15. We found that the following single point mutations in the switch-II region specifically attenuated the Arl15-Smad4 interaction: R90L, R95C, and Y96F, as demonstrated in our pull-down assays (*Figure 7e*). These mutations also substantially reduced the stimulation of Arl15-AL in the transcription of SBE ×4 luc under starvation (*Figure 7f*). A frameshift and several nonsense heterozygous mutations found in the cancer genomic database (*Figure 7—source data 4*) could downregulate cellular Arl15 protein level and reduce the normal TGFβ signaling activity, as shown by our Arl15 knockdown experiment (*Figure 6b*; *Figure 6—figure supplement 1d*). The TGFβ signaling pathway is tumor-suppressive for the proliferation of pre-malignant cancer cells (*Hao et al., 2019*; *Lamouille et al., 2014*; *Massagué, 2008*; *Seoane and Gomis, 2017*). Combined with cancer mutations in the Smad4-MH2 domain (*Figure 1j and k*), our data demonstrate that cancer patients can have genetic mutations or alterations that compromise the Arl15-Smad4 interaction and suggest that such genetic changes might contribute to tumorigenesis by down-regulating the TGFβ signaling pathway.

## Discussion

Our study uncovered Arl15 as a unique regulator of the TGFβ family signaling pathway. According to our knowledge, it is the first small G protein reported to interact with Smads. It positively regulates both TGFβ and BMP pathways by directly interacting with the MH2 domain of common Smad, Smad4, and promoting the assembly of the Smad-complex. Interestingly, the Smad-complex serves as an effector and a GAP of Arl15 so that it dissociates Arl15 and subsequently enters the nucleus.

Our work provides a molecular mechanism on how closed or autoinhibited Smad4 becomes open or activated, an outstanding question in the field (*Hata et al., 1997*). We discovered that opening and, hence, activation of Smad4 is specifically aided by Arl15-GTP through direct binding to the MH2 domain of Smad4. A small G protein is regulated by its GEFs and GAPs, activities of which could be subject to further intracellular or extracellular stimuli. Hence, our study reveals Arl15 as a potential signaling integration node between a small G-protein activation cascade and the TGFβ family signaling. The finding might provide a new clue to understanding the contextual and paradoxical nature of the TGFβ family signaling.

Based on our data, we propose a working model for the molecular role of Arl15 in the TGFβ family signaling (*Figure 7g*). A currently unidentified GEF, which is positively stimulated by TGFβ family cytokines, first activates Arl15-GDP to become GTP-loaded. Next, Smad4, mainly in a closed or inactive conformation, interacts with Arl15-GTP. Once bound to Arl15-GTP, Smad4-MH1 dissociates from its MH2 domain, rendering Smad4 in an open or active conformation. Although the Smad4-MH2 domain possesses a GAP activity toward Arl15, it might be too weak to inactivate Arl15-GTP in vivo. Therefore, a pool of Arl15-GTP-Smad4 complex might accumulate intracellularly. Upon stimulation by a

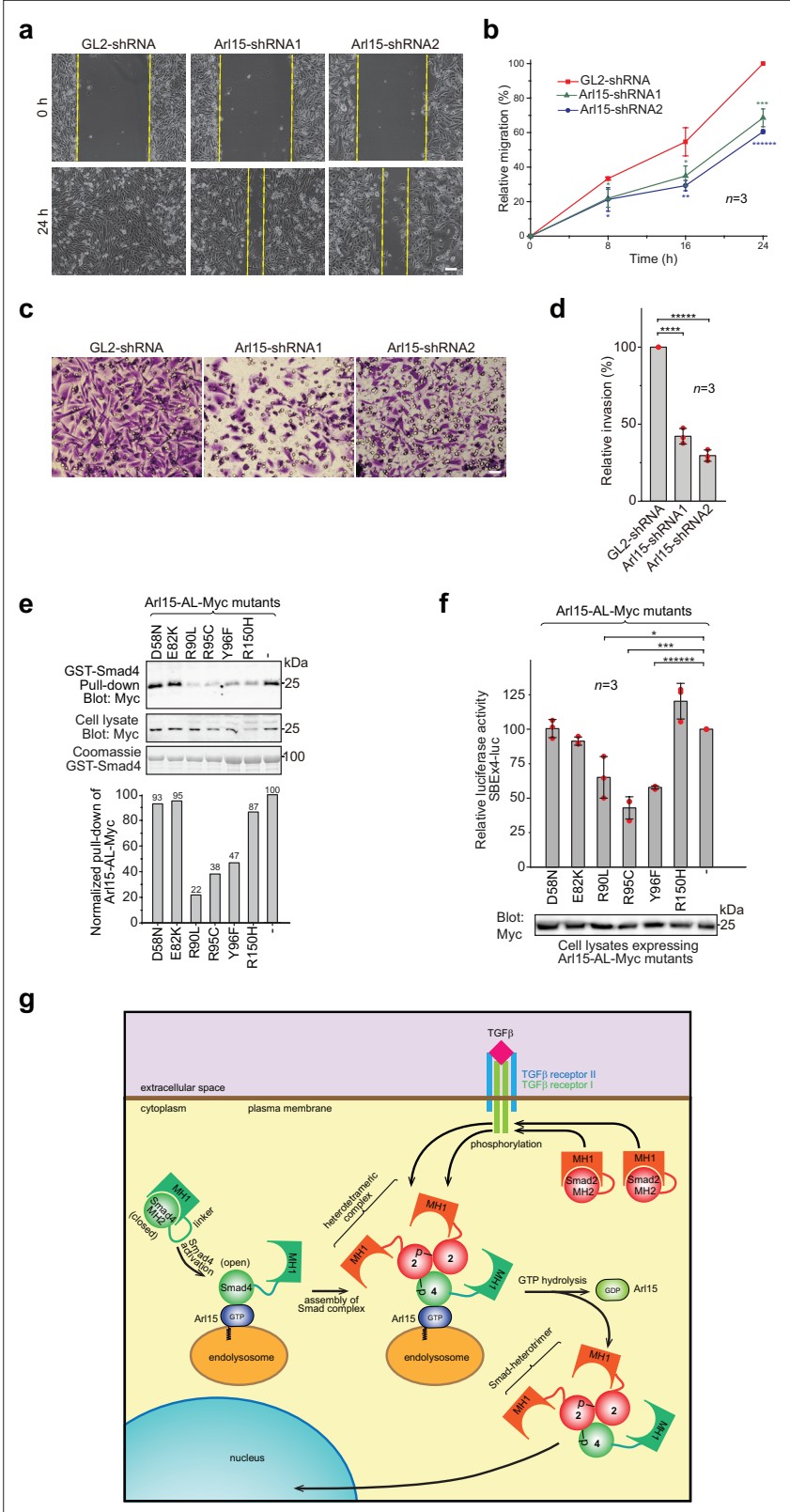

**Figure 7.** Implication of Arl15 in tumorigenesis by in vitro assays and mutation analysis, and our working model on how Arl15 regulates the TGFβ family signaling pathway. (**a, b**) Arl15 is required for the efficient in vitro migration of MDA-MB-231 cells. MDA-MB-231 cells were subjected to lentivirus-transduced knockdown using indicated shRNA. When cells reached confluency, a strip of cells was scraped off, and the resulting gap was live-imaged to

*Figure 7 continued on next page*

*Figure 7 continued*

monitor the migration of cells. The percentage of the relative migration (see Materials and methods) is plotted in (**b**). (**c,d**) Arl15 is required for the efficient in vitro invasion and migration of MDA-MB-231 cells. MDA-MB-231 cells were subjected to lentivirus-transduced knockdown using indicated shRNA and were subsequently placed into cell culture filter chambers with basement matrix. Cells that invaded through the matrix and migrated to the lower surface of the filter were stained in (**c**). In (**d**), the relative invasion was calculated as described in Materials and methods. Scale bar, 100 µm. (**e, f**) Arl15 missense mutations identified from cancer patients compromise Arl15-Smad4 interaction and TGFβ signaling. In (**e**), ead-immobilized GST-Smad4 was incubated with HeLa cell lysates expressing Arl15-AL-Myc harboring indicated mutation, and pull-downs and the cell lysates were immunoblotted against Myc-tag. The normalized pull-down of Arl15-AL-Myc is shown below, and it is calculated as the ratio of the pull-down band intensity to the corresponding cell lysate band intensity. Loading of fusion proteins is shown by Coomassie staining. In (**f**), HeLa cells were co-transfected to express the dual-luciferase and Arl15-AL-Myc with indicated mutation. After 20 hr starvation, cells were subjected to the dual-luciferase assay and Western blot analysis for Myc-tag. In (**b**, **d**, and **f**), error bar, mean ± SD of n=3 experiments. p values were from the *t*-test (unpaired and two-tailed). *, p≤0.05; **, p≤0.005; ***, p≤0.0005; ******, p≤0.0000005. Red dot, individual data point. Molecular weights (in kDa) are labeled in all immunoblots. (**g**) A working model illustrating the molecular mechanism on how Arl15 regulates the TGFβ family signaling pathway. Smad2 is used as an example of the R-Smad. 2, Smad2; 4, Smad4; p, phosphate group at Smad2. See text for details.

The online version of this article includes the following source data and figure supplement(s) for figure 7:

**Source data 1.** Uncropped gel and blot images for *Figure 7*.

**Source data 2.** Numerical data for graphs in *Figure 7b, d and f*.

**Source data 3.** List of ARL15 missense cancer mutations that are tested in GST-Smad4 pull-down assay (*Figure 7e and f*).

**Source data 4.** List of ARL15 frameshift and nonsense mutations from COSMIC.

**Figure supplement 1.** Arl15 missense mutations identified in cancer patients do not affect the GTP-binding of Arl15-AL.

**Figure supplement 1—source data 1.** Uncropped gel and blot images for *Figure 7—figure supplement 1*.

---

TGFβ family cytokine, R-Smads (Smad2 as an example in *Figure 7g*) are phosphorylated by TGFβ type I receptor kinase at their C-termini. With an activated MH2 domain, Arl15-bound Smad4 subsequently interacts with phospho-R-Smads. Since the R-Smad and Smad4 are known to assemble as a hetero-trimer (*Chacko et al., 2001*; *Chacko et al., 2004*), it is tempting to speculate that one Arl15-GTP and one Smad-heterotrimer form a heterotetramer. We further assume that associated Arl15 probably inhibits the nuclear translocation of the Smad-complex. By engaging R-Smads, the GAP activity of Smad4 is greatly enhanced and consequently triggers the GTP hydrolysis of Arl15. After the resulting Arl15-GDP dissociates from the Smad-complex, the latter translocates to the nucleus and executes eventual genomic actions.

Our model proposes that the Smad-complex is an effector and a terminator for active Arl15, which promotes the assembly of the Smad-complex in the first place. A similar case has been proposed for Sar1 and COPII coat complex (*Antonny et al., 2001*; *Bi et al., 2002*; *Bi et al., 2007*). Sar1 is an Arf-family small G protein that initiates the assembly of the COPII coat at the ER exit site. The COPII coat consists of repetitive units of two heterodimers, Sec23-Sec24, and Sec13-Sec31. In Sec23-Sec24 heterodimer, Sec23 functions as both an effector and a GAP for Sar1. The GAP activity of Sec23 is substantially boosted after Sec23-Sec24 recruits Sec13-Sec31 to complete the budding of a COPII-coated vesicle. The hydrolysis of Sar1-bound GTP eventually uncoats COPII-coated vesicles, which prepares them for the subsequent transport and fusion. We note that Arl15 and Sar1 have Ala (at 86) and His (at 77) in their G3-motifs, respectively, in contrast to Gln of other Arf-family small G proteins. Conserved Gln in the G3-motif functions to facilitate the GTP hydrolysis by correctly positioning the water molecule. Therefore, termination of active Sar1 and Arl15 in vivo probably requires their effector-cum-GAPs, which ensure the proper assembly of the COPII coat and the Smad-complex, respectively, before the complete dissociation of small G proteins. This property probably makes Sar1 and Arl15 a suitable initiator in promoting the assembly of large protein complexes. Confirmation of our speculation awaits future structural study of the Arl15-Smad-complex.

Our observation that Arl15 is required for the efficient invasion and migration of malignant cancer cells suggests that it might be a pro-metastatic factor. However, the finding of mutations compromising

the Arl15-Smad4 interaction in the cancer genomic database suggests that Al15 might be a tumor suppressor. It is well known that the TGFβ signaling pathway possesses a context-dependent dual-role in cancer progression (*Massagué, 2008*; *Massagué, 2012*). In normal epithelia or pre-malignant cancers, the TGFβ signaling pathway exerts a cytostatic or tumor-suppressive effect to inhibit the proliferation of cells. On the other hand, in malignant or metastatic cancers, TGFβ signaling exerts a pro-metastatic effect to promote growth and invasion of cells. An example is the effector of Arl15, Smad4 (*Deckers et al., 2006*). Consistent with the tumor-suppressive function of Smad4, depletion of Smad4 promotes the growth of NMuMG mammary gland epithelial cells; in contrast, the depletion of Smad4 inhibits the metastasis of MDA-MB-231 cells, in agreement with the pro-metastatic role of Smad4. Furthermore, inactivating mutations are commonly found in the *SMAD4* gene in pre-malignant cancers such as pancreatic carcinoma, while *SMAD4* has been reported as an essential gene for metastasis of breast cancer (*Levy and Hill, 2006*; *Massagué, 2008*; *Massagué, 2012*). Therefore, we hypothesize that, like Smad4, Arl15 might play a similar dual-role in cancer progression.

Although our study mainly focused on the TGFβ signaling pathway, we observed that Arl15 promotes the assembly of the BMP Smad-complex (*Figure 4—figure supplement 1*), and it is essential for the full activity of BMP R-Smad-dependent transcription (*Figure 6—figure supplement 1d*). Therefore, our data indicate that Arl15 also participates in the BMP signaling pathway. Recent evidence suggests that TGFβ and BMP pathways play opposite roles in animal development and diseases (*Ning et al., 2019*). It has been proposed that competition between TGFβ and BMP pathways for Smad4 contributes to their antagonistic cellular effects (*Candia et al., 1997*; *Sartori et al., 2013*; *Yuan et al., 2018*). Hence, we speculate that, similar to Smad4, Arl15 might mediate antagonistic crosstalk between the two pathways.

Genetic studies have implicated the *ARL15* gene locus in rheumatoid arthritis and multiple metabolic diseases (*Corre et al., 2018*; *Danila et al., 2013*; *Glessner et al., 2010*; *Li et al., 2014*; *Negi et al., 2013*; *Mahajan et al., 2014*; *Richards et al., 2009*; *Ried et al., 2016*; *Sun et al., 2015*; *Willer et al., 2013*). Consistent with the broad role of the TGFβ signaling pathway in physiology and pathology, our findings raise the likelihood of *ARL15* as the causative gene and suggest the contribution of the TGFβ family signaling pathway to the pathology of these metabolic diseases. We hypothesize that the expression level of *ARL15* might vary in particular genetic background, therefore correspondingly changing the strength of the TGFβ family signaling. Indeed, we found that the overexpression and depletion of Arl15 substantially modulate the TGFβ-dependent transcription (*Figure 6a–c and e*; *Figure 6—figure supplement 1b, d ,j ,k*). Therefore, our findings warrant a further investigation of the role of Arl15 in metabolic diseases.

## Materials and methods
### DNA plasmids
See *Supplementary file 1*.

### Antibodies, TGFβ family cytokines, and chemicals
Mouse anti-Smad2/3 mAb (#610842; 1:1000 for western blot or WB), mouse anti-GM130 mAb (#610823, 1:500 for immunofluorescence or IF) and mouse anti-golgin245 mAb (#611280, 1:200 for IF) were purchased from BD Bioscience. Rabbit anti-phospho-Smad2 (S465/467)/Smad3 (S423/425) mAb (#8828; 1:1000 for WB, 1: 200 for IF) and rabbit anti-Flag mAb (#14793; 1:1000 for WB) were purchased from Cell Signaling Technology. Rabbit anti-Nup133 mAb (#ab155990; 1:1000 for WB) and rabbit anti-glyceraldehyde 3-phosphate dehydrogenase (GAPDH) (#ab9485, 1:1000 for WB) were purchased from Abcam. The following antibodies were from Santa Cruz: mouse anti-Smad4 mAb (B-8, #sc-7966,1:1000 for WB), rabbit anti-Smad1/5/8 polyclonal antibody (pAb) (#sc-6031-R, 1:1000 for WB), rabbit anti-GAPDH pAb (#sc-25778, 1:1000 for WB), mouse anti-GFP mAb (#sc-9996, 1:1000 for WB, 1: 200 for IF), mouse anti-His mAb (#sc-8036, 1:1000 for WB), mouse anti-Myc mAb (#sc-40, 1:1000 for WB, 1:200 for IF) and mouse anti-HA mAb (#sc-7396, 1:1000 for WB). HRP (horseradish peroxidase)-conjugated goat anti-mouse (#176516, 1:10,000 for WB) and anti-rabbit IgG antibodies (#176515, 1:10,000 for WB) were from Bio-Rad. Alexa Fluor conjugated goat anti-mouse (1:500 for IF), anti-rabbit IgG antibodies (1:500 for IF) and recombinant human BMP2 (#PHC7145) were from Thermo Fisher Scientific. Recombinant human TGFβ1 (#100–21 C-10) was purchased from PeproTech.

A total of 20 µg/ml stock solution of TGFβ1 was made in 4 mM HCl containing 1 mg/ml bovine serum albumin. SB431542 (#1614) and guanosine 5'-[β,γ-imido]triphosphate (GMPPNP, #G0635) were purchased from Tocris and Sigma-Aldrich, respectively.

## Yeast-two hybrid screening

The yeast two-hybrid screening was conducted using the MatchMaker yeast two-hybrid system (Clontech) according to the company's manual. Gal4 DNA-binding domain fused Arl15-AL, constructed in pGBKT7 (Clontech), was used as a bait to screen Gal4 DNA-activation domain fused human kidney cDNA library (Clontech) by yeast mating, similar to previously described (*Lu et al., 2001*). DNA plasmids of positive hits were subsequently recovered, and prey cDNAs were identified by DNA sequencing.

## Cell culture and transfection

HeLa, HEK293T, MCF7, and MDA-MB-231 cells were from American Type Culture Collection. 293 FT cells were from Thermo Fisher Scientific. MCF7, MDA-MB-231, HeLa, HEK293T, and 293 FT cells were maintained in high glucose Dulbecco's Modified Eagle's Medium (DMEM) supplemented with 10% fetal bovine serum (FBS) (Thermo Fisher Scientific) at 37 °C in a 5% $CO_2$ incubator. The identity of all cell lines was authenticated by Short Tandem Repeat profiling (American Type Culture Collection). Cell lines were tested negative for mycoplasma contamination by DNA staining. FBS was heat-inactivated at 55 °C for 30 min. Cells were transfected using polyethylenimine (Polysciences) or Lipofectamine 2000 transfection reagent (Thermo Fisher Scientific).

During live-cell imaging, transfected HeLa cells grown on a glass-bottom Petri-dish (MatTek Corporation) were imaged in the $CO_2$-independent medium (Thermo Fisher Scientific) supplemented with 4 mM glutamine and 10% FBS at 37 °C.

## Knockdown and expression

293 FT cells grown in a 6-well plate were transfected with shRNA constructs in pLKO.1 vector or pLVX expression constructs together with pLP1, pLP2, and pLP/VSVG using Lipofectamine 2000 (Thermo Fisher Scientific). Eighteen hr after transfection, cells were incubated with fresh medium (DMEM supplemented with 10% FBS) at 37 °C for another 24–48 hr. The supernatant of the tissue culture medium was collected, passed through a 0.45 µm filter (Sartorius), and used immediately. For lentivirus-mediated knockdown or expression, cells were incubated for 24–48 hr with the virus supernatant supplemented with 8 µg/ml polybrene (Sigma-Aldrich #H9268) before being subjected to further experimental procedures.

ON-TARGETplus SMARTpool siRNA against human Smad4 (Smad4 siRNA) (#L-003902–00), a scrambled siRNA (control siRNA), and GL2 control siRNA were from Thermo Fisher Scientific. Smad2 (sense: Sense: 5'-GUCCCAUGAAAAGACUUAAtt-3' and anti-sense: 5'-UUAAGUCUUUUCAUGG GACtt-3') and Smad3 siRNAs (Sense: 5'-GGAGAAAUGGUGCGAGAAGtt-3' and anti-sense: 5'-CUUC UCGCACCAUUUCUCCtc-3')(*Phanish et al., 2006*) were synthesized by Thermo Fisher Scientific. Mon2-shRNA#2 was based on the Mon2 siRNA#2 described previously (*Mahajan et al., 2019*). Cells were transfected with siRNAs using Lipofectamine 2000 for two days before assays.

## Preparation of recombinant proteins

DNA plasmids encoding recombinant proteins were used to transform BL21 *E coli* bacterial cells. After induction by isopropyl β-D-1-thiogalactopyranoside, a bacterial cell pellet was collected. For purification of GST-fusion proteins (those cloned in pGEB or pGEX-KG vectors), cells were lysed using the freeze-thaw method in a buffer containing 50 mM Tris pH 8.0, 150 mM NaCl, 2 mM dithiothreitol (DTT), 1 mg/ml lysozyme, and 1 mM phenylmethylsulfonyl fluoride (PMSF). The cell lysate was centrifuged at 20,000 g for 30 min, and the resulting supernatant was incubated with Glutathione Sepharose 4B beads (GE Healthcare) overnight at 4 °C. The beads were washed three times with a buffer containing 50 mM Tris pH 8.0, 0.1% Triton-X 100, 150 mM NaCl, and 2 mM DTT. The bound proteins were eluted using reduced glutathione, and the resulting eluent was subjected to extensive dialysis against phosphate-buffered saline (PBS).

For purification of His-tagged proteins (those cloned in pET30a and pET30ax vectors), bacterial cells were lysed with a buffer containing 20 mM Tris pH 8.0, 150 mM NaCl, 10 mM imidazole,

2 mM DTT, 1 mg/ml lysozyme, and 1 mM PMSF and the resulting supernatant was incubated with nickel-nitrilotriacetic acid agarose (QIAGEN) overnight at 4 °C. The beads were washed with a buffer containing 20 mM Tris pH 8.0, 150 mM NaCl, 20 mM imidazole, and 2 mM DTT and eluted with an elution buffer containing 20 mM Tris pH 8.0, 150 mM NaCl, 250 mM imidazole, and 2 mM DTT. The eluent was subjected to extensive dialysis against PBS. All purified proteins were quantified by Coomassie staining in SDS-PAGE.

## Production of Arl15 antibody

DNA plasmid, His-Arl15-WT in pET30ax, was used to transform BL21 *E coli* cells. After induction by isopropyl β-D-1-thiogalactopyranoside, the bacterial cell pellet was lysed by sonication in PBS containing 8 M urea. After high-speed centrifugation, the supernatant was incubated with nickel-nitrilotriacetic acid agarose beads at room temperature for 2 hr. The beads were subsequently washed in PBS containing 8 M urea and 20 mM imidazole. Next, bead-bound His-Arl15-WT was eluted in PBS containing 8 M urea and 250 mM imidazole. After concentrating and changing the buffer to PBS containing 4 M urea, His-Arl15-WT was sent to Genemed Synthesis Inc for rabbit immunization and antiserum collection. To purify the polyclonal antibody against Arl15, GST-Arl15 immobilized on gluta-thione Sepharose beads was incubated with 50 mM dimethyl pimelimidate (Thermo Fisher Scientific) in 200 mM sodium borate pH 9.0 to crosslink GST-Arl15 covalently onto glutathione Sepharose beads. The crosslinked beads were subsequently washed with 200 mM ethanolamine pH 8.0, incubated with the antiserum at room temperature for 1 hr, and washed with PBS. Finally, the bound antibody was eluted by 100 mM glycine pH 2.8 and dialyzed against PBS.

## Immunoprecipitation and GST pull-down

HeLa cells were subjected to lentivirus-mediated knockdown using Arl15-shRNA1, Arl15-shRNA2, or GL2-shRNA. After that, cells were serum-starved for 4 hr followed by either further starvation or 5 ng/ml TGFβ1 treatment for 20 hr before cell lysis. HEK293T cells were transfected by indicated DNA constructs or siRNAs before cell lysis. Both types of cells were lysed with the lysis buffer (40 mM HEPES pH 7.4, 150 mM NaCl, 1% Trition X-100, 2.5 mM $MgCl_2$, 1×cOmplete Protease Inhibitor Cocktail (Roche), 1 mM PMSF, and 1 mM DTT). For immunoprecipitation, after centrifugation, the supernatant was incubated with indicated antibodies overnight at 4 °C. The antibody-antigen complex was captured by incubating with lysis buffer prewashed proteinA/G beads (Thermo Fisher Scientific) for 2 hr. In some immunoprecipitations, GFP-Trap beads (ChromoTek) were used to directly immuno-precipitate GFP-tagged fusion proteins. For GST pull-down, cleared cell lysate was incubated with bead-immobilized GST-fusion protein for 4–14 hr at 4 °C. The beads were subsequently washed with the lysis buffer. Next, the bound protein was eluted by boiling in SDS-sample buffer and resolved in the SDS-PAGE. SDS-PAGE separated proteins were transferred to polyvinyl difluoride membrane (Bio-Rad), which was sequentially incubated with the primary and HRP-conjugated secondary antibody. At last, the chemiluminescence signal was detected by a cooled charge-coupled device using LAS-4000 (GE Healthcare Life Sciences). Alternatively, the chemiluminescence signal was detected by CL-XPo-sure film (Thermo Fisher Scientific) and digitally scanned. Uncropped gel and blot images are shown as source data.

## GTP-agarose pull-down

Transiently transfected HEK293T cells were suspended in a binding buffer (20 mM HEPES, pH7.4, 150 mM NaCl, 1×cOmplete Protease Inhibitor Cocktail). After three cycles of freeze-thaw and extru-sion through a 25 ½ gauge needle, the cell lysate was cleared by centrifugation at 16,000 g for 30 min and subsequently treated with 5 mM EDTA (final concentration) for 1 hr at 4 °C. Next, the lysate was incubated with GTP-agarose beads (bioWORLD) for 1 hr in the presence of 10 mM $MgCl_2$ (final concentration) at 4 °C. After washing with the binding buffer four times, beads were boiled in the SDS-sample buffer and analyzed by Western blot.

## Immunofluorescence

Cells grown on Φ12 mm glass coverslips were fixed with 4% paraformaldehyde in PBS at room temperature for 20 min and washed with 100 mM ammonium chloride and PBS. Next, cells were sequentially incubated with primary and fluorescence-conjugated secondary antibodies, which were

diluted in the fluorescence dilution buffer (PBS supplemented with 5% FBS and 2% bovine serum albumin) containing 0.1% saponin (Sigma-Aldrich). After extensive washing with PBS, coverslips were mounted in the Mowiol mounting medium, containing 12% Mowiol 4–88 (EMD Millipore), 30% glycerol, and 100 mM Tris pH 8.5.

## Wide-field fluorescence microscopy

Unless specified, all fluorescence images were acquired under an inverted wide-field fluorescence microscope (Olympus IX83) equipped with a Plan Apo oil objective lens (63×or 100×oil, NA 1.40), a motorized stage, motorized filter cubes, a scientific complementary metal oxide semiconductor camera (Neo; Andor Technology), and a 200 watt metal-halide excitation light source (LumenPro 200; Prior Scientific). Dichroic mirrors and filters in filter cubes were optimized for Alexa Fluor 488/GFP, 594/mCherry and 647. The microscope system was controlled by MetaMorph software (Molecular Devices), and only the center quadrant of the camera sensor was used for imaging. During the live-cell imaging, HeLa cells grown on a glass-bottom Petri-dish (as described above in Cell culture and transfection) were imaged in a 37 °C chamber.

## Laser scanning confocal microscopy

HeLa cells co-expressing mCherry and GFP-tagged proteins were grown on a glass-bottom Petri-dish as described above (Cell culture and transfection). Live-cell imaging was conducted in a 37 °C chamber under Zeiss LSM710 laser scanning confocal microscope system (Carl Zeiss) equipped with a Plan-apochromat objective (100 x oil, NA 1.40). Two laser lines with wavelengths of 488 nm and 561 nm were used to excite GFP and mCherry, and their emission filter bandwidths were 495–550 nm and 595–620 nm, respectively. The microscope system was controlled by ZEN software (Carl Zeiss).

## Dual-luciferase assay

HeLa cells cultured in 24-well plates were transfected with indicated firefly luciferase reporter construct (SBE ×4 luc or BRE-luc) together with pRL-SV40 renilla luciferase control vector (Promega) using Lipofectamine 2000 (Thermo Fisher Scientific). Constant amount of total transfected DNA was balanced by supplying pBluescript SK vector DNA to the transfection mixture. Twenty-four hr after transfection, cells were serum-starved for 4 hr and treated with 5 ng/ml TGFβ1 for 20 hr. Cells were subsequently lysed, and firefly and renilla luciferase activities were measured using the Dual-Luciferase Reporter Assay System (Promega) according to standard protocol.

## RT-qPCR

Total RNA was extracted from MCF7 or MDA-MB-231 cells using Trizol reagent (Thermo Fisher Scientific) according to the manufacturer's protocol. Reverse transcription primed by random nonamer primers was conducted using nanoScript 2 Reverse Transcription kit (Primerdesign). The qPCR was performed using SYBR green-based PrecisionFAST kit (Primerdesign) in Bio-Rad CFX96 Touch real-time PCR detection system. Melt curves and agarose gel electrophoresis were performed to confirm the specificity of PCR primers. The qPCR result of each gene was first divided by that of β-tubulin and further normalized to that of control (empty vector or GL2-shRNA treatment). Primers for qPCR are as follows: c-Myc (5'-AAAGGCCCCCAAGGTAGTTA-3'; 5'-GCACAAGAGTTCCGTAGCTG-3'), ID1 (5'-CAAATTCAAGGTGGAATCGAA-3'; 5'-GGTGGCTGGGAAGTGAACTA-3'), $p21^{cip1}$ (5'-GAGGCCGG GATGAGTTGGGAGGAG-3'; 5'-CAGCCGGCGTTTGGAGTGGTAGAA-3'), $p27^{kip1}$ (5'-GCTCCACA GAACCGGCATTT-3'; 5'-AAGCGACCTGCAACCGACGATTCTT-3'), E-cadherin (5'-TCTTCCCCGCCC TGCCAATC-3'; 5'-GCCTCTCTCGAGTCCCCTAG-3'), N-cadherin (5'-GGTGGAGGAGAAGAAGACCA G-3'; 5'-GGCATCAGGCTCCACAGT-3'), α-SMA (5'-CTATGCCTCTGGACGCACAACT-3'; 5'-CAGA TCCAGACGCATGATGGCA-3'), Fibronectin (5'-ACAACACCGAGGTGACTGAGAC-3'; 5'-GGAC ACAACGATGCTTCCTGAG-3'), β-tubulin (5'-TTGGCCAGATCTTTAGACCAGACAAC-3'; 5'-CCGT ACCACATCCAGGACAGAATC-3') and Arl15 (5'-CCCCGATAACGTCGTGTC-3'; 5'-AGCGGCTCCAGT ATTTCC-3').

There were some modifications in the experiment described in *Figure 6e*. Lentivirus was harvested by transfecting pMD2.G, psPAX2 together with Arl15-shRNA1, Arl15-shRNA2 or GL2-shRNA in pLKO.1. After lentivirus infection, pooled MCF7 cells were transiently selected with puromycin. The resulting cells were treated with 5 ng/ml TGFβ1 for 72 hr and total RNA was extracted by using TRIzol

reagent and Qiagen RNeasy Mini Kit (Qiagen) according to the manufacturer's protocol. Reverse transcription primed by random hexamer primer was conducted using RevertAid H Minus First Strand cDNA Synthesis Kit (Thermo Fisher Scientific). SensiFAST SYBR Hi-ROX Kit (Bioline) in QuantStudio 5 (Thermo Fisher Scientific). The qPCR result of each gene was first divided by that of ribosomal protein L13A mRNA (Primers: 5'-GCC TTC ACA GCG TAC GA-3'; 5'-CGA AGA TGG CGG AGG TG-3') and further normalized to that of GL2-shRNA control.

## Nuclear fractionation

HeLa cells were first subjected to lentivirus-mediated knockdown using Arl115-shRNA2 or GL2-shRNA. Subsequently, cells were serum-starved for 4 hr followed by either further starvation or 5 ng/ml TGFβ1 treatment for 20 hr. Next, cells collected by scraping culture flasks were washed three times with ice-cold PBS. After centrifugation at 200 g for 5 min, pelleted cells were re-suspended in 500 µl buffer A (20 mM Tris pH 7.4, 10 mM NaCl, 3 mM $MgCl_2$, 0.5 % NP40, 1 mM DTT and 1 mM PMSF) and incubated on ice for 15 min. Cells were subsequently vortexed for 10 s, and the resulting cell lysate was centrifuged for 10 min at 1000 g at 4 °C. The supernatant, which the non-nuclear or cytoplasmic fraction, was transferred into a new tube, while the pellet, which is the nuclear fraction, was washed three times using buffer A without NP40. Both fractions were subjected to the SDS-PAGE and western blot analysis.

## Image analysis

All image analysis was conducted in ImageJ (http://imagej.nih.gov/ij/).

## GAP assay

A similar protocol has been previously described (*Pan et al., 2006*). Briefly, purified His-Arl15-WT and His-Arl15-AL proteins were incubated with a 20-fold molar excess of GTP in 20 mM HEPES pH7.5, 150 mM NaCl, 5 mM ethylenediaminetetraacetic acid, and 1 mM DTT at room temperature for 1 hr. Next, the proteins were subjected to a 7 KDa molecular weight cut-off Zeba Spin Desalting Column (Thermo Fisher Scientific), which was pre-equilibrated with 20 mM HEPES pH7.5 and 150 mM NaCl. The GAP assay was conducted in a 96-well glass-bottom microplate (Corning), and the released inorganic phosphate was measured using EnzChek Phosphate Assay Kit (Thermo Fisher Scientific). The reaction system contained 20 mM HEPES pH 7.5, 150 mM NaCl, 0.15 mM 2-amino-6-mercapto-7-m ethylpurine ribonucleoside, 0.75 U/ml purine nucleoside phosphorylase, 10 mM $MgCl_2$, 40 µM GTP-loaded His-Arl15-WT or His-Arl15-AL, and 0.4 µM following GAP candidate proteins, single or in combinations as indicated in text: His-Smad4, His-Smad2-SE, GST-Smad4, GST-Smad4-MH1, GST-Smad4-linker-MH2, GST-Smad4-MH2, and GST (negative control). The kinetics of the GTP hydrolysis was continuously monitored by the absorbance at 360 nm in Cytation 5 (BioTek) at 22 °C. For each time series, absorbance values were subtracted by the corresponding initial value measured at 0 min.

## ER-to-Golgi and Golgi export trafficking assays

These assays were performed as previously described (*Mahajan et al., 2019*). HeLa cells subjected to lentivirus-transduced shRNA knockdown were further transfected to express a RUSH reporter: Ii-Strep_ManII-SBP-GFP or Ii-Strep_TNFα-SBP-GFP (*Boncompain et al., 2012*) for the ER-to-Golgi or Golgi export to the PM transport assay. Fifty ng/ml streptavidin was added to cell culture and was removed 20 hr after transfection of RUSH reporters. For the ER-to-Golgi trafficking assay, 40 µM biotin and 10 µg/ml cycloheximide were added to the cell medium during the chase. For the Golgi export trafficking assay, cells were first incubated at 20 °C for 3 hr in the presence of 40 µM biotin and 10 µg/ml cycloheximide to accumulate TNFα-SBP-GFP at the Golgi. The system was subsequently warmed up to 37 °C during the chase. In both assays, cells were processed for immunofluorescence at various chase times and imaged by the wide-field microscope. The Golgi fraction of RUSH reporter is calculated using $I_{Golgi}/I_{cell}$, in which $I_{Golgi}$ and $I_{cell}$ are integrated GFP intensity of the Golgi and the cell, respectively. All cells positively expressing RUSH reporter were analyzed in each image.

## Cancer mutation data of Arl15

Cancer mutation data of Arl15 were manually compiled from cBioPortal (https://www.cbioportal.org/) and COSMIC (https://cancer.sanger.ac.uk/cosmic/gene/analysis?ln=ARL15).

## Wound healing migration assay

MDA-MB-231 cells were first subjected to lentivirus-transduced knockdown using GL2-shRNA, Arl15-shRNA1, or 2. Next, cells were cultured to confluence in 6-well plates. Gaps were scratched across a well using a pipette tip, and the closure of gaps was kinetically monitored by an inverted phase contrast microscope. The widths of gaps were quantified using ImageJ (https://imagej.nih.gov/ij/). The percentage of the relative migration was calculated as $(1-d/d_0)*100\%$, in which $d_0$ is the initial width of the gap and $d$ is the width of the gap at a specific time.

## Invasion assay

MDA-MB-231 cells were first subjected to lentivirus-transduced knockdown using GL2, Arl15-shRNA1, or 2. After cells were serum-starved for 24 h in DMEM, the same amount of suspended cells were added into upper chambers of Corning Costar Transwell cell culture inserts (pore size 8 µm; Sigma-Aldrich, #CLS3464) with the coating of 30 µg Matrigel Basement Membrane Matrix (BD Biosciences). A total of 700 µl DMEM supplemented with 10% FBS was added into the lower chamber. After 24 hr incubation at 37 °C in a $CO_2$ incubator, the upper chamber was washed three times with PBS, fixed with ice-cold methanol, and stained with crystal violet solution (0.5% crystal violet in 20% methanol). Finally, cells on the upper surface of the insert were removed with a cotton swab, and those on the lower surface (translocated cells) were imaged. In each experiment, five randomly selected fields were imaged, and the number of cells within each image was counted and averaged. The relative invasion was calculated as the number of cells per image normalized by that of control (GL2-shRNA treated).

## Acknowledgements

We thank Z Ding (Temasek Polytechnic, Singapore) for the help in the yeast-two hybrid screening and R Derynck (University of California San Francisco, USA), P ten Dijke (Leids Universitair Medisch Centrum, Netherlands), W Hong (Institute of Molecular and Cell Biology, Singapore), T Kirchhausen (Harvard Medical School, USA), M Lowe (University of Manchester, UK), F Perez (Institut Curie, France), D Trono (EPFL, Switzerland), D Root (Broad Institute, USA), M Roussel (St. Jude Children's Research Hospital, USA) for sharing DNA plasmids. This work was supported by the following grants to LL.: MOE AcRF Tier1 RG35/17, Tier2 MOE2015-T2-2-073, and MOE2018-T2-2-026.

## Additional information

### Funding

| Funder | Grant reference number | Author |
|---|---|---|
| Ministry of Education - Singapore | AcRF Tier1 RG35/17 | Meng Shi<br>Hieng Chiong Tie<br>Mahajan Divyanshu<br>Xiuping Sun<br>Yan Zhou<br>Boon Kim Boh<br>Lei Lu |
| Ministry of Education - Singapore | Tier2 MOE2015-T2-2-073 | Meng Shi<br>Hieng Chiong Tie<br>Mahajan Divyanshu<br>Xiuping Sun<br>Yan Zhou<br>Boon Kim Boh<br>Lei Lu |
| Ministry of Education - Singapore | MOE2018-T2-2-026 | Meng Shi<br>Hieng Chiong Tie<br>Mahajan Divyanshu<br>Xiuping Sun<br>Yan Zhou<br>Boon Kim Boh<br>Lei Lu |

| Funder | Grant reference number | Author |
| --- | --- | --- |

The funders had no role in study design, data collection and interpretation, or the decision to submit the work for publication.

## Author contributions

Meng Shi, Data curation, Formal analysis, Investigation, Methodology, Writing – review and editing; Hieng Chiong Tie, Mahajan Divyanshu, Data curation, Formal analysis, Investigation; Xiuping Sun, Yan Zhou, Boon Kim Boh, Data curation, Formal analysis; Leah A Vardy, Supervision, Project administration; Lei Lu, Conceptualization, Resources, Formal analysis, Supervision, Funding acquisition, Investigation, Methodology, Writing – original draft, Project administration, Writing – review and editing

## Author ORCIDs

Meng Shi ⓘ http://orcid.org/0000-0002-8119-9757
Hieng Chiong Tie ⓘ http://orcid.org/0000-0003-2738-8685
Lei Lu ⓘ http://orcid.org/0000-0002-8192-1471

## Decision letter and Author response

Decision letter https://doi.org/10.7554/eLife.76146.sa1
Author response https://doi.org/10.7554/eLife.76146.sa2

# Additional files

## Supplementary files

• Supplementary file 1. List of DNA plasmids used in this study.

• Transparent reporting form

## Data availability

All data generated or analyzed during this study are included in the manuscript and supporting file. Source Data files have been provided.

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
