## [Editor Report]

In their manuscript, Shi et al. provide intriguing evidence for a novel regulatory mechanism of signaling in the transforming growth factor-β pathway and they identify through a two-hybrid screen a novel interactor of Smad4 called Arl15, which is a small G-protein. They demonstrate that Arl15 binds the MH2 domain of Smad4 leading to a release of its autoinhibitory structure which facilitates complex assembly with the R-Smads. An important mechanistic feature of this study is that the authors show these interactions are induced upon transforming growth factor-β stimulation. Finally, consistent with the proposed mechanism the authors show that a limited set of known downstream transcription factor targets of the transforming growth factor-β pathway are dependent upon Arl15.

---

## [Decision Letter]

**Decision letter after peer review:**

Thank you for submitting your article "Arl15 upregulates the TGFβ family signaling by promoting the assembly of the Smad-complex" for consideration by *eLife*. Your article has been reviewed by 2 peer reviewers, one of whom is a member of our Board of Reviewing Editors, and the evaluation has been overseen by Philip Cole as the Senior Editor. The reviewers have opted to remain anonymous.

The reviewers felt that your results were intriguing but concern was raised regarding the physiological significance of your findings. In particular, based upon the comments from the reviewers it will be essential in the revised manuscript that additional experimentation be provided to address the issue of whether the Arl15 signaling mechanism is TGF-b responsive. Also, whether the reported interactions, particularly between Arl15 and Smad 4 with Smad 2/3 occur at endogenous levels in a regulated TGF-b manner. Additionally, it will be essential to demonstrate that Arl15 modulates Smad 2/3 phosphorylation in response to TGF-b, again under endogenous conditions. We feel that these experiments are key to supporting the physiological relevance of your findings and strongly encourage you to directly address these issues with further experimentation.

*Reviewer #1 (Recommendations for the authors):*

1. The localization data in figure 1 seems misplaced as an opening figure. That Arl15 is localized to a number of different membranes including the Golgi and plasma membrane is of interest but the presentation of these data as Figure 1 is distracting and disrupts the flow of the ensuing data as the relevance is vague. Although one could argue that Arl15-AL mutant localization data showing that it is localized to the plasma membrane would raise technical concerns over how the two-hybrid experiment was able to identify Smad4 as an interactor. The data in Figure 1 neither setup nor provide any rationale for the following two-hybrid screen that was performed with Arl15 as bait.

2. The interaction data is convincing although the authors do not show whether the interaction occurs at endogenous levels of both Smad4 and Arl15. Also, of importance is the question of whether this interaction is responsive to TGF-β activation.

3. Figure 2 is somewhat inconclusive since the authors are comparing the puncta staining "intensity" between full-length Smad4 and the MH2 domain alone. This difference in intensity could simply be a result of the MH2 domain alone expressing to much higher levels than that of full-length Smad4. This is very likely to be the case. The authors should modify their interpretation of these results or provide some control expression data for Smad4 in this experiment.

4. Can the authors add a table in the supplementary section listing the interactors identified in the two hybrid screen. What other partners were found?

5. Discrete point mutants in the MH2 and switch II domains of Smad4 and Arl15, respectively would provide stronger evidence for the specificity of the interaction and further mitigate any possibility that the failure to interact through these regions is simply due to inappropriate protein folding.

6. Figure 4D which shows quantitation of the previous pull-downs – this table should be labeled with the actual experimental conditions instead of just lanes 1-4.

7. It is surprising that there is a stoichiometric dissociation of Arl15 from Smad4. The authors should perform biochemical fractionation experiments with Arl15 in the presence and absence of TGF-β to substantiate their claim that Arl15 fails to piggy-back with Smad4 to the nucleus.

8. What is the status of Smad2/3 phosphorylation in response to TGF-β when Arl15 is knocked down.

9. The TGF-β pathway plays an important role also in the activation of fibrogenic genes, are these repertoire of targets (e.g. collagens and fibronectins) also affected by Arl15 loss-of-function/gain-of-function mutants.

10. In the Results section the authors simply refer to cancer-associated mutants from "cancer patients" the type of cancer that these mutants have been identified in, the frequency of occurrence should be provided.

*Reviewer #2 (Recommendations for the authors):*

I previously reviewed this manuscript for another journal. The figure set has remained identical and there are modest changes in wording. This very unfortunate, considering how much effort goes into reviewing a manuscript with the aim to improve it. My review is therefore nearly identical to my previous one.

– Line 105-106: The statement that "we found that … the MH1 domain and linker region is insufficient to interact…" is not supported by Figure 1e. The linker region is not tested.

– Figure 1g, h: Only Arl15-AL and -TN are tested. What is the interaction with wild-type Arl15?

– Section starting on line 128: The colocalization of Smad4 with Arl15 is not convincing.

– Lines 149-155: Related to Figure 3b, the right panel is compared with the left panel, but it is unclear if the Smad band intensities of these two panels can be compared with each other, especially since it is unclear if these two panels derive from one gel with one exposure intensity.

– Related to the same experiment(s), does repressing endogenous Smad4 levels using shRNA result in decreased R-Smad interaction?

– Line 165: The concept of "activation" of Smad4 is unclear and unprecedented to my knowledge. Can one really say that decreased MH1-MH2 domain interaction confers activation?

– Figure 4c: This panel needs to additionally show the retention of HA-Smad4-MH2 by GST-Smad2-MH1 to allow for better interpretation of the data.

– Section starting line 184: This section needs to show the effects of wt Arl15 and Arl15-AL on the TGF-b-induced association of endogenous Smad2 or Smad3 with endogenous Smad4, and on the BMP-induced association of endogenous Smad1 or Smad5 with endogenous Smad4. This is key!

– Additionally, this section needs to show the effects of wt Arl15 and Arl15-AL on the TGF-b-induced nuclear translocation of endogenous Smad2 or Smad3 (with endogenous Smad4), and on the BMP-induced nuclear of endogenous Smad1 or Smad5 with endogenous Smad4. Immunofluorescence may be insufficiently informative, but nuclear versus cytosolic fractionation should visualize any effects.

– Data in Figure 6c: What is the effect of wt Arl15 (not only Arl15-AL)?

– Considering that Arl15-AL enhances the TGF-b-induced transcription response of direct target genes, does Arl15-AL enhance the TGF-b-induced binding of Smad complexes to regulatory gene sequences, as assessed by ChIP? Does silencing Arl15 expression decrease the binding of these complexes?

---

## [Author Response]

The reviewers felt that your results were intriguing but concern was raised regarding the physiological significance of your findings. In particular, based upon the comments from the reviewers it will be essential in the revised manuscript that additional experimentation be provided to address the issue of whether the Arl15 signaling mechanism is TGF-b responsive. Also, whether the reported interactions, particularly between Arl15 and Smad 4 with Smad 2/3 occur at endogenous levels in a regulated TGF-b manner. Additionally, it will be essential to demonstrate that Arl15 modulates Smad 2/3 phosphorylation in response to TGF-b, again under endogenous conditions. We feel that these experiments are key to supporting the physiological relevance of your findings and strongly encourage you to directly address these issues with further experimentation.

We have conducted new experiments to address these concerns raised by reviewers.

Regarding "whether the Arl15 signaling mechanism is TGF-b responsive", we demonstrated that TGFβ1 treatment increased the interaction between Arl15 and Smad4. See our response to point 2 of Reviewer #1's comments.

Regarding “whether interactions of "Arl15 and Smad4 with Smad 2/3 occur at endogenous levels in a TGF-b regulated manner", we conducted new co-IP assays using endogenous proteins and demonstrated the followings. (1) TGFβ1 treatment increased the interaction between Arl15 and Smad4 (see our response to point 2 of Reviewer #1's comments). (2) Arl15 promotes the assembly of the Smad-complex induced by the TGFβ1 treatment (see our response to point 8 of Reviewer #2's comments). (3) Smad4 is essential to mediate the indirect interaction between endogenous Arl15 and phospho-Smad2/3 (see our response to point 5 of Reviewer #2's comments).

Regarding whether "Arl15 modulates Smad 2/3 phosphorylation in response to TGF-b", our data show that depletion of Arl15 does not affect the TGFβ1-stimulated phosphorylation of Smad2/3 (see our response to point 8 of Reviewer #1's comments).

Reviewer #1 (Recommendations for the authors):1. The localization data in figure 1 seems misplaced as an opening figure. That Arl15 is localized to a number of different membranes including the Golgi and plasma membrane is of interest but the presentation of these data as Figure 1 is distracting and disrupts the flow of the ensuing data as the relevance is vague. Although one could argue that Arl15-AL mutant localization data showing that it is localized to the plasma membrane would raise technical concerns over how the two-hybrid experiment was able to identify Smad4 as an interactor. The data in Figure 1 neither setup nor provide any rationale for the following two-hybrid screen that was performed with Arl15 as bait.

We have modified our manuscript as suggested by this reviewer. We moved the Golgi and plasma membrane (PM) localization data of Arl15 from Figure 1 and SFigure 1 to Figure 2 and Figure 2—figure supplement 1, respectively. The corresponding text was also modified accordingly.

2. The interaction data is convincing although the authors do not show whether the interaction occurs at endogenous levels of both Smad4 and Arl15. Also, of importance is the question of whether this interaction is responsive to TGF-β activation.

As suggested by this reviewer, we tested the endogenous interaction between Smad4 and Arl15 and explored if TGFβ1 regulates the interaction. Indeed, we confirmed the interaction between endogenous Smad4 and Arl15. Furthermore, we found that TGFβ1 stimulates their interaction. The new data have been incorporated into Figure 1.

The following sentences have been added to the Results.

“Next, we asked if the TGFβ1 treatment regulates the interaction between endogenous Smad4 and Arl15. We found that, under the control shRNA knockdown (GL2-shRNA), TGFβ1 treatment significantly increased the amount of Arl15 IPed by endogenous Smad4 (Figure 1c and d). Therefore, our finding suggests that TGFβ1 could stimulate the interaction between Smad4 and Arl15 through a currently unknown mechanism.”

3. Figure 2 is somewhat inconclusive since the authors are comparing the puncta staining "intensity" between full-length Smad4 and the MH2 domain alone. This difference in intensity could simply be a result of the MH2 domain alone expressing to much higher levels than that of full-length Smad4. This is very likely to be the case. The authors should modify their interpretation of these results or provide some control expression data for Smad4 in this experiment.

We think that this reviewer is referring to Figure 2c and d (previous Figure 2 a,b).

We did not intend to compare the intensity of the puncta. Instead, we wanted to compare the ratio of the punctate to the cytosolic background intensity. The difference in the two ratios is visually obvious in Figure 2 c and d — the ratio of the MH2-domain is much higher than that of the full-length.

We have modified the corresponding text to avoid confusion. We now use "barely discernable", instead of weak punctate appearance, to describe the low ratio of the punctate to the cytosolic background intensity.

“Although immunostaining did not reveal a clear membrane association of Smad4, our live-cell confocal imaging uncovered a limited colocalization between mCherry-Smad4 and Arl15-AL-GFP at punctate structures (Figure 2c). Notably, the punctate appearance of mCherry-Smad4 was barely discernable against its cytosolic pool. The poor punctate localization might be due to the closed conformation of Smad4, formed by the intramolecular interaction between its MH1 and MH2 domains (see below).”

4. Can the authors add a table in the supplementary section listing the interactors identified in the two hybrid screen. What other partners were found?

As suggested, we prepared the below table as Figure 1- table supplement 1.

5. Discrete point mutants in the MH2 and switch II domains of Smad4 and Arl15, respectively would provide stronger evidence for the specificity of the interaction and further mitigate any possibility that the failure to interact through these regions is simply due to inappropriate protein folding.

We have tested six mutations in the switch-II region of Arl15 in Figure 7e and f.

We also screened eight missense cancer mutations in the Smad4-MH2 domain to search for essential amino acids for the interaction of Smad4 with Arl15-GTP. We found that A532D, E538K, and H541Y of Smad4 might be involved in binding to Arl15-GTP using pull-down assays.

The below text has been added to the Results.

“To search for amino acids in the Smad4-MH2 domain essential for interacting with Arl15, we screened eight missense cancer mutations within the Smad4-MH2 domain (Figure 1-table supplement 2). Briefly, we prepared GFP-tagged full-length Smad4 with single-point mutations and tested their interactions with Arl15-AL in the GST pull-down assay. We found that mutations, A532D, E538K, and H541Y, but not M447K, D493H, L495P, R496H, and R497H, substantially disrupted the pull-down of GFP-Smad4 by GST-Arl15-AL (Figure 1j and k), suggesting that the region from amino acid 532 to 541 of Smad4 might be involved in the interaction with Arl15-GTP.”

6. Figure 4D which shows quantitation of the previous pull-downs – this table should be labeled with the actual experimental conditions instead of just lanes 1-4.

We have modified the figure panels as suggested.

7. It is surprising that there is a stoichiometric dissociation of Arl15 from Smad4. The authors should perform biochemical fractionation experiments with Arl15 in the presence and absence of TGF-β to substantiate their claim that Arl15 fails to piggy-back with Smad4 to the nucleus.

The requested experiment has already been shown in Figure 5b of our previous manuscript. We have performed a more comprehensive new experiment to replace the previous Figure 5b. In both experiments, red boxes indicate the absence of Arl15 in the nuclear fractions under starvation or TGFβ1 treatment.

8. What is the status of Smad2/3 phosphorylation in response to TGF-β when Arl15 is knocked down.

Our previous (Supplementary Figure 6i or Figure 6—figure supplement 1i) and new data (blot in Figure 1c and its quantification in Figure 4h) demonstrated that the TGFβ1-stimulated phosphorylation of Smad2/3 is not affected by the depletion of Arl15.

9. The TGF-β pathway plays an important role also in the activation of fibrogenic genes, are these repertoire of targets (e.g. collagens and fibronectins) also affected by Arl15 loss-of-function/gain-of-function mutants.

We measured the transcriptional change of Fibronection and α-SMA genes under the overexpression of Arl15-AL and TN in MCF7 cells. We did not analyze collagen since its expression was undetectable in MCF7 cells. We found that overexpression of Arl15-AL, but not TN or the empty vector, upregulates the transcription of Fibronection and α-SMA genes, consistent with the positive role of Arl15 in TGFβ signaling. The new data are now shown in Figure 6d.

The corresponding text has been revised below.

“The increase of p27^kip1^ and p21^cip1^ and decrease of c-Myc transcripts correlate with the cytostatic effect of TGFβ signaling, while the rise of N-cadherin, Snail1, Fibronectin, and α-SMA, and reduction of E-cadherin transcripts characterize the EMT (Hao et al., 2019; Lamouille et al., 2014; Seoane and Gomis, 2017; Xu et al., 2009). On the other hand, under the TGFβ1 treatment, depletion of Arl15 in MCF7 cells reversed the transcriptional trend of the above genes compared to control knockdown, i.e., those that were upregulated by Arl15-AL overexpression (e.g., N-cadherin, ID1, Snail1, p27^kip1^, and p21^cip1^) became downregulated, and vice versa (e.g., E-cadherin and c-Myc) (Figure 6e).”

10. In the Results section the authors simply refer to cancer-associated mutants from "cancer patients" the type of cancer that these mutants have been identified in, the frequency of occurrence should be provided.

We have prepared the following tables as suggested by this reviewer. Figure 1-table supplement 2, Figure 7-table supplement 1 and Figure 7-table supplement 2

Reviewer #2 (Recommendations for the authors):I previously reviewed this manuscript for another journal. The figure set has remained identical and there are modest changes in wording. This very unfortunate, considering how much effort goes into reviewing a manuscript with the aim to improve it. My review is therefore nearly identical to my previous one.– Line 105-106: The statement that "we found that … the MH1 domain and linker region is insufficient to interact…" is not supported by Figure 1e. The linker region is not tested.

We have deleted "linker region" in the corresponding text. Below is the text after modification.

“We found that the Smad4-MH2 domain, but not the MH1 domain, is sufficient to interact with Arl15-GTP.”

– Figure 1g, h: Only Arl15-AL and -TN are tested. What is the interaction with wild-type Arl15?

Figure 1b (previously Figure 1d) shows that endogenous Smad4 can interact with endogenous Arl15 in the presence of GMPPNP. Furthermore, as suggested by this reviewer, we also employed co-IPs to confirm the interaction between endogenous Arl15 and Smad4 under starvation and TGFβ1 stimulation. Our new data are shown in Figure 1 c and d.

– Section starting on line 128: The colocalization of Smad4 with Arl15 is not convincing.

It is unclear why the colocalization between Smad4 and Arl15 is not convincing to this reviewer. We think that this reviewer refers to the colocalization between full-length Smad4 and Arl15-AL. Indeed, full-length Smad4 does not have a good membrane localization at all, possibly due to its closed conformation. However, the colocalization between Smad4-MH2 and Arl15-AL should be clear.

– Lines 149-155: Related to Figure 3b, the right panel is compared with the left panel, but it is unclear if the Smad band intensities of these two panels can be compared with each other, especially since it is unclear if these two panels derive from one gel with one exposure intensity.

1) The left (without the expression of GFP-Smad4) and right panels (with the expression of GFP-Smad4) were from different gels. Figure 3 – source data 1 are their corresponding uncropped gel images. Therefore, each gel has its own 1% cell lysate input lane for the purpose of quantification in Figure 3c.

2) We did not directly compare the band intensities between the left and right Western blot panels of Figure 3b. Instead, we calculated in Figure 3c the percentage of Flag-Smad1 pulled down in Figure 3b using the 1% cell lysate input lane. From the plot of Figure 3c, the percentage of Flag-Smad1 pulled down showed a significant increase with the expression of GFP-Smad4.

– Related to the same experiment(s), does repressing endogenous Smad4 levels using shRNA result in decreased R-Smad interaction?

We performed the experiment as advised by this reviewer. As expected, under TGFβ1 treatment, we found that Smad4 depletion significantly reduced the amount of phospho-Smad2/3 IPed by Arl15-AL-GFP. Figure 3d, e, and f show our new data.

– Line 165: The concept of "activation" of Smad4 is unclear and unprecedented to my knowledge. Can one really say that decreased MH1-MH2 domain interaction confers activation?

This activation of Smad4 is a new concept proposed in our manuscript. The following three pieces of evidence strongly suggest such activation by Arl15-GTP.

1) The interaction between Arl15-GTP and the Smad4-MH2 domain decreases the intra-molecular interaction between Smad4-MH1 and -MH2 domains (Figure 4a and b), resulting in an open conformation in Smad4.

2) Arl15-GTP engaged Smad4-MH2 interacts with the MH2 domain of R-Smads more robustly (Figure 4c and d).

3) Figure 4 e, f, h, and i, and Figure 4—figure supplement 1 demonstrate that Arl15-GTP could increase the interaction between Smad4 and phospho-R-Smad.

– Figure 4c: This panel needs to additionally show the retention of HA-Smad4-MH2 by GST-Smad2-MH1 to allow for better interpretation of the data.

Smad4-MH2 does not interact with Smad2-MH1 (Hata *et al.*, Nature, 1997). Therefore, we expect that no HA-Smad4-MH2 is retained by GST-Smad2-MH1.

References

Hata A, Lo RS, Wotton D, Lagna G, Massagué J. Mutations increasing autoinhibition inactivate tumour suppressors Smad2 and Smad4. Nature. 1997 Jul 3;388(6637):82-7. doi: 10.1038/40424. PMID: 9214507.

– Section starting line 184: This section needs to show the effects of wt Arl15 and Arl15-AL on the TGF-b-induced association of endogenous Smad2 or Smad3 with endogenous Smad4, and on the BMP-induced association of endogenous Smad1 or Smad5 with endogenous Smad4. This is key!

As suggested by this reviewer, we tested the interaction between endogenous Smad4 and phospho-Smad2/3, which was stimulated by TGFβ1 treatment, upon the depletion of Arl15 (see Figure 1c and Figure 4h and i). We found that depletion of Arl15 reduces the interaction between Smad4 and Phospho-Smad2/3, therefore suggesting that Arl15 might be essential for efficient Smad-complex assembly. We have incorporated these new results into Figure 1c and Figure 4h and i.

We have added the following text in the Results section of our manuscript.

“To study Arl15's effect on the Smad-complex assembly in the context of endogenous proteins, we investigated the interaction between Smad4 and phospho-Smad2/3 upon the depletion of Arl15 (Figure 1c). We first demonstrated that depleting Arl15 does not significantly change the cellular level of phospho-Smad2/3 stimulated by the TGFβ1 treatment (Figure 4h). Importantly, consistent with our above overexpression studies, we found that the depletion of Arl15 substantially reduced phospho-Smad2/3 IPed by Smad4 (Figure 4i), suggesting an essential role of Arl15 in efficient Smad-complex assembly.”

– Additionally, this section needs to show the effects of wt Arl15 and Arl15-AL on the TGF-b-induced nuclear translocation of endogenous Smad2 or Smad3 (with endogenous Smad4), and on the BMP-induced nuclear of endogenous Smad1 or Smad5 with endogenous Smad4. Immunofluorescence may be insufficiently informative, but nuclear versus cytosolic fractionation should visualize any effects.

As advised by this reviewer, we tested the TGFβ1-induced nuclear translocation of endogenous phospho-Smad2/3, Smad4, and Arl15 upon depletion of Arl15 and overexpression of Arl15-AL using the nucleus fractionation assay (see Figure 5 b-d and g-i).

1) We found that Arl15 was undetectable in the nuclear fraction under starvation or TGFβ1 treatment in control knockdown cells. However, in control and Arl15 knockdown cells, nuclear fractions of phospho-Smad2/3 and Smad4 increased substantially under the TGFβ1 treatment compared to the starvation treatment. Importantly, when Arl15 was depleted, substantially less phospho-Smad2/3 and Smad4 were detected in the nuclear fraction (Figure 5 b-d). Our finding thus supports our hypothesis that Arl15 is essential for the efficient assembly of the Smad-complex, which ensures the subsequent nuclear translocation of the Smad-complex.

We have modified or added the following text in the Results part of our manuscript.

“We then asked if Arl15-GTP co-translocates to the nucleus together with the Smad-complex. In fluorescence imaging, we observed that Arl15 did not localize to the nucleus under the TGFβ1 treatment (Figure 5a). In contrast, the nuclear localization of phospho-Smad2/3 increased substantially (Figure 5a). We then further confirm our imaging result and explore the role of Arl15 in the nuclear localization of the Smad-complex using the nuclear fractionation assay. Consistent with our imaging, we did not detect Arl15 in the nuclear fraction in starved or TGFβ1 stimulated control knockdown cells (Figure 5b). However, in control or Arl15 knockdown cells, nuclear fractions of phospho-Smad2/3 and Smad4 substantially increased under the TGFβ1 treatment compared to the starvation treatment (Figure 5b). Similar observations were made for Arl15 and phospho-Smad1/5/8 in BMP2 stimulated cells (Figure 5—figure supplement 1a). Our findings are consistent with our current knowledge of the TGFβ family signaling pathway, in which phospho-R-Smads and Smad4 translocate to the nucleus upon stimulation by TGFβ family cytokines (Derynck and Budi, 2019; Massague, 2012; Schmierer and Hill, 2007; Wrana, 2013). Notably, when Arl15 was depleted, substantially less phospho-Smad2/3 and Smad4 were found in the nuclear fraction (Figure 5 b-d). The finding thus supports our hypothesis that Arl15 is essential for the assembly of the Smad-complex, which subsequently translocates to the nucleus.”

2) We observed that overexpression of Arl15-AL decreased the nuclear fraction of phospho-Smad2/3 and Smad4 compared to empty vector control, suggesting that bound Arl15-AL inhibits the nuclear translocation of the Smad-complex. We have modified or added the following text in the Results part of our manuscript.

“To better understand the significance of dissociating Arl15 from the Smad-complex, we investigated the effect of overexpressing Arl15-AL on the TGFβ1-stimulated nuclear translocation of the Smad-complex. Since Arl15-AL does not significantly hydrolyze the bound GTP (Figure 5e and f), we expect that it could tightly bind to the Smad-complex with minimal dissociation. HeLa cells transfected with Arl15-AL or empty vector (control) were subjected to starvation or TGFβ1-stimulation as Figure 5b. Our nuclear fractionation assay demonstrates that Arl15-AL reduced the percentage of nuclear Smad4 and phospho-Smad2/3 compared to the control (Figure 5g-i). Therefore, when bound, Arl15 seems to inhibit the nuclear translocation of the Smad-complex. The Smad-complex must inactivate and dissociate the bound Arl15 for efficient nuclear translocation.”

– Data in Figure 6c: What is the effect of wt Arl15 (not only Arl15-AL)?

The effect of Arl15-WT should be weaker than Arl15-AL since not all Arl15-WT is in the GTP-binding or active form. This prediction is supported by experiments in Figure 6a and Figure 6—figure supplement 1b. Under the starvation treatment, Arl15-WT promotes the transcription of TGFβ1 and BMP pathway luciferase reporters more weakly than Arl15-AL.

– Considering that Arl15-AL enhances the TGF-b-induced transcription response of direct target genes, does Arl15-AL enhance the TGF-b-induced binding of Smad complexes to regulatory gene sequences, as assessed by ChIP? Does silencing Arl15 expression decrease the binding of these complexes?

Arl15 promotes the assembly and nuclear translocation of the Smad-complex. The increased nuclear concentration of Smad-complex is expected to increase the binding of the Smad-complex to the DNA sequences of target genes, as reported by extensive literature. However, Arl15 does not follow the Smad-complex to the nucleus, and it remains in the cytoplasm. Our study focuses on Arl15 and its role in assembling the Smad-complex. Therefore, we think that the investigation of Smad-complex's DNA binding in the nucleus is not within the scope of the current study. Instead, it should belong to the future study.